# Uncertainty-aware off policy learning

## Abstract

Off-policy learning, referring to the procedure of policy optimization with access only to logged feedback data, has shown importance in various real-world applications, such as search engines, recommender systems, etc. While the ground-truth logging policy, which generates the logged data, is usually unknown, previous work directly takes its estimated value in off-policy learning, resulting in a biased estimator. This estimator has both high bias and variance on samples with small and inaccurate estimated logging probabilities. In this work, we explicitly model the uncertainty in the estimated logging policy and propose a novel Uncertainty-aware Inverse Propensity Score estimator (UIPS) for improved off-policy learning. Experiment results on synthetic and three real-world recommendation datasets demonstrate the advantageous sample efficiency of the proposed UIPS estimator.

## 1 Introduction

In many real-world applications, including search engines (Agarwal et al. (2019)), online advertisements (Strehl et al. (2010)), recommender systems (Chen et al. (2019); Liu et al. (2022)), only logged feedback data is available for subsequent policy optimization. For example, in recommender systems, various complex recommendation models (i.e., policies) (Zhou et al. (2018); Guo et al. (2017)) were optimized with logged user interactions (e.g., clicks or staytime) to items recommended by previous recommendation policies. However, such logged data is known to be biased, since one does not know the feedback on items that previous policy (which is generally referred as the *logging policy*) did not take. This inevitably distorts the evaluation and optimization of a new policy when it tends to select items that are not in the logged data.

Off-policy learning (Thrun & Littman (2000); Precup (2000)) emerges as a favorable way to learn an improved policy only from the logged data by addressing the mismatch between the learning policy and the logging policy. One of the most commonly used off-policy learning methods is Inverse Propensity Scoring (IPS) (Chen et al. (2019); Munos et al. (2016)), which assigns per-sample importance weight to the training objective on the logged data, so as to get an unbiased optimization objective in expectation. The importance weight in IPS is the probability ratio between the learning policy and the logging policy.

However, the ground-truth logging policy is unavailable to the learner, e.g., it is not recorded in the data. One common treatment taken by previous work (Strehl et al. (2010); Liu et al. (2022); Chen et al. (2019); Ma et al. (2020)) is to first employ a supervised learning method (e.g., logistic regression, neural networks, etc.) to estimate the logging policy, and then take the estimated logging policy for off-policy learning. We theoretically show that such an approximation results in a biased estimator which is sensitive to those inaccurate and small estimated logging probabilities. Worse still, the small values of the estimated logging probabilities usually mean that there are fewer related samples in the logged data, so its estimation usually has high uncertainties, i.e., inaccurate estimation with high probability. Figure 1 shows a piece of empirical evidence from a large-scale recommendation benchmark KuaiRec dataset (Gao et al. (2022)), where items with lower frequencies in the logged dataset have lower estimated logging probabilities and higher uncertainties concurrently. The high bias and variance caused by these samples greatly hinder the performance of off-policy learning.

In this work, we explicitly take the uncertainty of the estimated logging policy into consideration and design a novel Ucertainty-aware Inverse Propensity Score estimator (UIPS) as the optimization objective for policy learning. UIPS introduces an additional weight to approach the ground-truth propensity from the estimated one, and learns an improved policy by alternating: (1) Find the optimal weight that makes the estimator as accurate as possible, taking into consideration the uncertainty

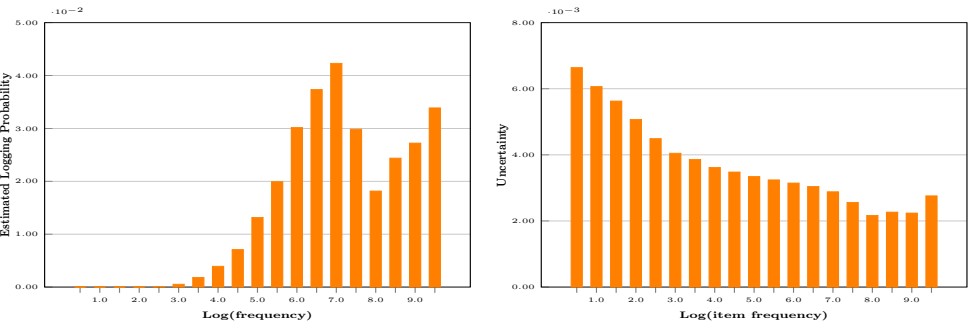

(a) Estimated Logging Probability        (b) Uncertainty of Estimation

Figure 1: Estimated logging policy and its uncertainty under different item frequency on KuaiRec.

of the estimated logging policy; (2) Improve the policy by optimizing the resulting estimator. We further find a closed-form solution for the optimal weight by deriving an upper bound on the mean squared error (MSE) to the ground-truth policy value. The optimal weight adjusts sample weights considering both the uncertainty of estimated logging probabilities and the propensity scores, rather than simply boosting or penalizing samples with high uncertain logging probabilities. Experiment results on the synthetic and three real-world recommendation datasets demonstrate the efficiency of UIPS. All data and code can be found in supplementary materials for reproducibility.

To summarize, our contribution in this work is as follows:

- We point out that directly using the estimated logging policy leads to sub-optimal off-policy learning, since the resulting biased estimator is greatly distorted by samples with inaccurate and small estimated logging probabilities.
- We take the uncertainty of the estimated logging policy into consideration and propose UIPS for more accurate off-policy learning.
- Experiments on synthetic and three real-world recommendation datasets demonstrate UIPS's strong advantage over state-of-the-art methods.

## 2    PRELIMINARY: OFF-POLICY LEARNING

We focus on the standard contextual bandit setup to explain the key concepts. Following convention (Joachims et al. (2018); Saito & Joachims (2022); Su et al. (2020)), let $\boldsymbol{x} \in \mathcal{X} \subseteq R^d$ be a $d$-dimensional context vector drawn from an unknown probability distribution $p(\boldsymbol{x})$. Each context is associated with a finite set of actions denoted by $\mathcal{A}$, where $|\mathcal{A}| < \infty$. Let $\pi : \mathcal{A} \times \mathcal{X} \to [0, 1]$ denote a stochastic policy, such that $\pi(a|\boldsymbol{x})$ is the probability of selecting action $a$ under context $\boldsymbol{x}$ and $\sum_{a \in \mathcal{A}} \pi(a|\boldsymbol{x}) = 1$. Under a given context, reward $r_{\boldsymbol{x},a}$ is observed when action $a$ is chosen. Take news recommendation for example, $\boldsymbol{x}$ represents the state of a user, summarizing his/her interaction history with the recommender system, each action $a$ is a candidate news article, the policy is a recommendation algorithm, and the reward $r_{\boldsymbol{x},a}$ denotes the user feedback on article $a$, e.g., whether the user clicks the article. Let $V(\pi)$ denote the expected reward or value of the policy $\pi$:

$$V(\pi) = \mathbb{E}_{\boldsymbol{x} \sim p(\boldsymbol{x}), a \sim \pi(a|\boldsymbol{x})}[r_{\boldsymbol{x},a}]. \tag{1}$$

We look for a policy $\pi(a|\boldsymbol{x})$ to maximize $V(\pi)$. In the rest of the paper, we denote $\mathbb{E}_{\boldsymbol{x} \sim p(\boldsymbol{x}), a \sim \pi(a|\boldsymbol{x})}[\cdot]$ as $\mathbb{E}_\pi[\cdot]$ for simplicity.

In contrast to performing online updates by following the learning policy $\pi(a|\boldsymbol{x})$, in off-policy learning we can only access a set of logged feedback data denoted by $D := \{(\boldsymbol{x}_n, a_n, r_{\boldsymbol{x}_n, a_n}) | n \in [N]\}$, where $[N] := \{1, \ldots, N\}$. Given $\boldsymbol{x}_n$, the action $a_n$ was generated by a stochastic logging policy $\beta^*$, i.e., the probability action $a_n$ was selected is $\beta^*(a_n|\boldsymbol{x}_n)$. The actions $\{a_1, \ldots, a_N\}$ and their corresponding rewards $\{r_{\boldsymbol{x}_1, a_1}, \ldots, r_{\boldsymbol{x}_N, a_N}\}$ are generated independently given $\beta^*$. Due to the nature of policy optimization, the learning policy $\pi(a|\boldsymbol{x})$ is expected to be different from $\beta^*(a|\boldsymbol{x})$, unless $\beta^*(a|\boldsymbol{x})$ is already optimal. Moreover, in practice the situation could be further complicated. Again, consider the news recommendation scenario. Due to the scalability requirement, industrial recommender systems usually adopt a two-stage framework (Ma et al. (2020)), where one or several

candidate generation models first produce a candidate set and a separate ranking model reranks candidate items to present top-K item to users. While $\beta^*(a|\boldsymbol{x})$ depicts the whole two-stage process, the learning policy $\pi(a|\boldsymbol{x})$ is usually employed in one particular stage (e.g., the reranking stage), implying drastic differences between the logging and learning policies. The main challenge of off-policy learning is then to address the distribution discrepancy between $\beta^*(a|\boldsymbol{x})$ and $\pi(a|\boldsymbol{x})$, and learn a policy $\pi(a|\boldsymbol{x})$ to maximize $V(\pi)$ with access only to the logged dataset $D$.

One of most widely used methods to address the distribution shift between $\pi(a|\boldsymbol{x})$ and $\beta^*(a|\boldsymbol{x})$ is the Inverse Propensity Score (IPS) (Chen et al. (2019); Munos et al. (2016)). One can easily get that:

$$V(\pi) = \mathbb{E}_{\beta^*} \left[ \frac{\pi(a|\boldsymbol{x})}{\beta^*(a|\boldsymbol{x})} r_{\boldsymbol{x},a} \right],$$

yielding the following empirical estimator of $V(\pi)$:

$$\hat{V}_{\text{IPS}}(\pi) = \frac{1}{N} \sum_{n=1}^{N} \frac{\pi(a_n|\boldsymbol{x}_n)}{\beta^*(a_n|\boldsymbol{x}_n)} r_{\boldsymbol{x}_n,a_n}, \tag{2}$$

where $\pi(a_n|\boldsymbol{x}_n)/\beta^*(a_n|\boldsymbol{x}_n)$ is referred to as the propensity score. In the rest of paper, without further specification, we use the empirical estimation of expectation in our practical calculation. Various algorithms can be readily used for policy optimization under $\hat{V}_{\text{IPS}}(\pi)$, including value-based methods (Silver et al. (2016)), policy-based methods (Levine & Koltun (2013); Schulman et al. (2015); Williams (1992)). In this work, we adopt a well-known policy gradient algorithm, REINFORCE (Williams (1992)). Assume the policy $\pi(a|\boldsymbol{x})$ is parameterized by $\boldsymbol{\vartheta}$, via the "log-trick", the gradient of $\hat{V}_{\text{IPS}}(\pi_{\boldsymbol{\vartheta}})$ with respect to $\boldsymbol{\vartheta}$ can be readily derived as follows:

$$\nabla_{\boldsymbol{\vartheta}} \hat{V}_{\text{IPS}}(\pi_{\boldsymbol{\vartheta}}) = \frac{1}{N} \sum_{n=1}^{N} \frac{\pi(a_n|\boldsymbol{x}_n)}{\beta^*(a_n|\boldsymbol{x}_n)} r_{\boldsymbol{x}_n,a_n} \nabla_{\boldsymbol{\vartheta}} \log(\pi_{\boldsymbol{\vartheta}}(a_n|\boldsymbol{x}_n)). \tag{3}$$

**Approximation with unknown logging policy**. In many real-world applications, the ground-truth logging policy, i.e., the $\beta^*(a|\boldsymbol{x})$ of each observation $(\boldsymbol{x}, a)$, is unknown. One reason is the legacy issue, i.e., the probabilities were not logged when collecting data. Another reason is that the exact value of $\beta^*(a|\boldsymbol{x})$ is intrinsically unavailable such as in the two-stage recommender systems. As the solution, previous work employs various supervised learning methods (e.g., logistic regression (Schnabel et al. (2016)), nerural networks (Chen et al. (2019), etc.) to estimate the logging policy, and replaces $\beta^*(a|\boldsymbol{x})$ with its estimated value $\hat{\beta}(a|\boldsymbol{x})$ to get the following estimator for policy learning:

$$\hat{V}_{\text{BIPS}}(\pi_{\boldsymbol{\vartheta}}) = \frac{1}{N} \sum_{n=1}^{N} \frac{\pi_{\boldsymbol{\vartheta}}(a_n|\boldsymbol{x}_n)}{\hat{\beta}(a_n|\boldsymbol{x}_n)} r_{\boldsymbol{x}_n,a_n}. \tag{4}$$

However, as shown in the following proposition, inaccurate $\hat{\beta}(a|\boldsymbol{x})$ leads to high bias and variance of $\hat{V}_{\text{BIPS}}(\pi_{\boldsymbol{\vartheta}})$. Worse still, smaller inaccurate $\hat{\beta}(a|\boldsymbol{x})$ further enlarges this bias and variance.

**Proposition 1.** *The bias and variance of $\hat{V}_{\text{BIPS}}(\pi_{\boldsymbol{\vartheta}})$ can be derived as follows:*

$$\text{Bias}\left(\hat{\text{V}}_{\text{BIPS}}(\pi_{\boldsymbol{\vartheta}})\right) = \mathbb{E}_D\left[\hat{V}_{\text{BIPS}}(\pi_{\boldsymbol{\vartheta}}) - V(\pi_{\boldsymbol{\vartheta}})\right] = \mathbb{E}_{\pi_{\boldsymbol{\vartheta}}}\left[r_{\boldsymbol{x},a}\left(\frac{\beta^*(a|\boldsymbol{x})}{\hat{\beta}(a|\boldsymbol{x})} - 1\right)\right]$$

$$N \cdot \text{Var}_D\left(\hat{V}_{\text{BIPS}}(\pi_{\boldsymbol{\vartheta}})\right) = \text{Var}_{\pi_{\boldsymbol{\vartheta}}}\left(\frac{\beta^*(a|\boldsymbol{x})}{\hat{\beta}(a|\boldsymbol{x})} r_{\boldsymbol{x},a}\right) + \mathbb{E}_{\pi_{\boldsymbol{\vartheta}}}\left[\left(\frac{\pi_{\boldsymbol{\vartheta}}(a|\boldsymbol{x})}{\beta^*(a|\boldsymbol{x})} - 1\right) \cdot \frac{\beta^*(a|\boldsymbol{x})^2}{\hat{\beta}(a|\boldsymbol{x})^2} r_{\boldsymbol{x},a}^2\right]$$

Smaller $\hat{\beta}(a|\boldsymbol{x})$ usually implies fewer related training samples in the logged data, and thus $\hat{\beta}(a|\boldsymbol{x})$ will be inaccurate with a higher probability. To make it more explicit, we take KuaiRec dataset (Gao et al. (2022)) as an example and estimate the logging policy following (Chen et al. (2019)). Figure 1 shows the estimated $\hat{\beta}(a|\boldsymbol{x})$ and its corresponding uncertainties in items of different observation frequencies in the logged dataset. As uncertainty measures how large the confidence interval is about the current estimation, higher uncertainty implies that the true value may be away from the empirical mean estimate with a high probability. We defer the discussion about our detailed uncertainty calculation in Section 3. We can observe from Figure 1 that as item frequency decreases, the estimated logging probability also decreases, but the estimation uncertainty increases. This implies that smaller $\hat{\beta}(a|\boldsymbol{x})$ is usually 1) more inaccurate and 2) associated with high uncertainty.

As a result, with high bias and variance caused by inaccurate $\hat{\beta}(a|\boldsymbol{x})$, it is erroneous to improve $\pi_{\boldsymbol{\vartheta}}(a|\boldsymbol{x})$ by simply optimizing $\hat{V}_{\text{BIPS}}(\pi_{\boldsymbol{\vartheta}})$. We propose uncertainty-aware off-policy learning to address this challenge.

## 3 UNCERTAINTY-AWARE OFF-POLICY LEARNING

Our idea is incorporating the uncertainty of the logging policy estimation into policy learning. Observing that

$$V(\pi_{\boldsymbol{\vartheta}}) = \mathbb{E}_{\beta^*}\left[\frac{\pi_{\boldsymbol{\vartheta}}(a|\boldsymbol{x})}{\hat{\beta}(a|\boldsymbol{x})} \cdot \frac{\hat{\beta}(a|\boldsymbol{x})}{\beta^*(a|\boldsymbol{x})} \cdot r_{\boldsymbol{x},a}\right],$$

we propose to learn the optimal policy by optimizing the following empirical estimator:

$$\hat{V}_{\text{UIPS}}(\pi_{\boldsymbol{\vartheta}}) = \frac{1}{N}\sum_{n=1}^{N} \frac{\pi_{\boldsymbol{\vartheta}}(a_n|\boldsymbol{x}_n)}{\hat{\beta}(a_n|\boldsymbol{x}_n)} \cdot \phi_{\boldsymbol{x}_n,a_n} \cdot r_{\boldsymbol{x}_n,a_n} \tag{5}$$

where $\phi_{\boldsymbol{x}_n,a_n}$ is a weight, which reflects $\hat{\beta}(a_n|\boldsymbol{x}_n)/\beta^*(a_n|\boldsymbol{x}_n)$, to be selected to make $\hat{V}_{\text{UIPS}}(\pi_{\boldsymbol{\vartheta}})$ as close to $V(\pi_{\boldsymbol{\vartheta}})$ as possible. Intuitively, one should give small weights to samples whose $\hat{\beta}(a|\boldsymbol{x})$ is far below the ground-truth $\beta^*(a|\boldsymbol{x})$. Thus, we divide offline policy improvement into two steps, and repeat them until certain convergence condition is met:

- **Uncertainty aware policy evaluation**: Derive the optimal uncertainty aware $\phi_{\boldsymbol{x},a}$ to make $\hat{V}_{\text{UIPS}}(\pi_{\boldsymbol{\vartheta}})$ as accurate as possible.
- **Policy Improvement**: Learn an improved policy $\pi_{\boldsymbol{\vartheta}}(a|\boldsymbol{x})$ by optimizing $\hat{V}_{\text{UIPS}}(\pi_{\boldsymbol{\vartheta}})$.

### 3.1 Uncertainty Aware Policy Evaluation

**Optimal uncertainty aware weight $\phi_{\boldsymbol{x},a}$.** We measure the accuracy of $\hat{V}_{\text{UIPS}}(\pi_{\boldsymbol{\vartheta}})$ by its mean squared error (MSE) to $V(\pi_{\boldsymbol{\vartheta}})$ following previous work (Su et al. (2020); Saito & Joachims (2022)). MSE captures both the bias and variance of an estimator, since it is the summation of squared bias and variance. We then locate the $\phi_{\boldsymbol{x},a}$ that can minimize the MSE. In particular, we demonstrate the optimal $\phi_{\boldsymbol{x},a}$ has a closed-form formula which relates to both the value of $\pi_{\boldsymbol{\vartheta}}(a|\boldsymbol{x})/\hat{\beta}(a|\boldsymbol{x})$ and the estimation uncertainty of $\hat{\beta}(a|\boldsymbol{x})$.

More specifically, instead of directly minimizing the MSE, which is intractable, we find the desirable $\phi_{\boldsymbol{x},a}$ by minimizing the upper bound of MSE in the following theorem.

**Theorem 1.** *Assume $r_{\boldsymbol{x},a} \in [0,1]$, the mean squared error (MSE) between $\hat{V}_{\text{UIPS}}(\pi_{\boldsymbol{\vartheta}})$ and ground-truth estimator $V(\pi_{\boldsymbol{\vartheta}})$ is upper bounded as follows:*

$$\text{MSE}\left(\hat{V}_{\text{UIPS}}(\pi_{\boldsymbol{\vartheta}})\right) = \mathbb{E}_D\left[\left(\hat{V}_{\text{UIPS}}(\pi_{\boldsymbol{\vartheta}}) - V(\pi_{\boldsymbol{\vartheta}})\right)^2\right] = \text{Bias}\left(\hat{V}_{\text{UIPS}}(\pi_{\boldsymbol{\vartheta}})\right)^2 + \text{Var}\left(\hat{V}_{\text{UIPS}}(\pi_{\boldsymbol{\vartheta}})\right)$$

$$\leq \mathbb{E}_{\pi_{\boldsymbol{\vartheta}}}\left[r_{\boldsymbol{x},a}^2 \frac{\pi_{\boldsymbol{\vartheta}}(a|\boldsymbol{x})}{\beta^*(a|\boldsymbol{x})}\right] \cdot \mathbb{E}_{\beta^*}\left[\left(\frac{\beta^*(a|\boldsymbol{x})}{\hat{\beta}(a|\boldsymbol{x})}\phi_{\boldsymbol{x},a} - 1\right)^2\right] + \mathbb{E}_{\beta^*}\left[\frac{\pi_{\boldsymbol{\vartheta}}(a|\boldsymbol{x})^2}{\hat{\beta}(a|\boldsymbol{x})^2}\phi_{\boldsymbol{x},a}^2\right]$$

The upper bound in Theorem 1 strictly increases with the two expectations related to $\phi_{\boldsymbol{x},a}$, which implies that for some choice $\lambda \in [0,\infty]$, the MSE-optimizing $\phi_{\boldsymbol{x},a}$ can be derived by minimizing:

$$\lambda\mathbb{E}_{\beta^*}\left[\left(\frac{\beta^*(a|\boldsymbol{x})}{\hat{\beta}(a|\boldsymbol{x})}\phi_{\boldsymbol{x},a} - 1\right)^2\right] + \mathbb{E}_{\beta^*}\left[\frac{\pi_{\boldsymbol{\vartheta}}(a|\boldsymbol{x})^2}{\hat{\beta}(a|\boldsymbol{x})^2}\phi_{\boldsymbol{x},a}^2\right]. \tag{6}$$

We cannot directly minimize Eq.(6) since the unknown $\beta^*(a|\boldsymbol{x})$ is involved. However, various ways (Gal & Ghahramani (2016); Xu et al. (2021)) can be employed to get the confidence interval which will contain $\beta^*(a|\boldsymbol{x})$ with high probability. More specifically, following previous work (Joachims et al. (2018)), we assume $\beta^*(a|\boldsymbol{x})$ can be modelled by a softmax function on top of an unknown function $f_{\boldsymbol{\theta}^*}(\boldsymbol{x},a)$, i.e., the realizable assumption. Then we can get:

$$\beta^*(a|\boldsymbol{x}) = \frac{\exp(f_{\boldsymbol{\theta}^*}(\boldsymbol{x},a))}{\sum_{a'}\exp(f_{\boldsymbol{\theta}^*}(\boldsymbol{x},a'))}, \quad \hat{\beta}(a|\boldsymbol{x}) = \frac{\exp(f_{\boldsymbol{\theta}}(\boldsymbol{x},a))}{\sum_{a'}\exp(f_{\boldsymbol{\theta}}(\boldsymbol{x},a'))}, \tag{7}$$

where $f_{\boldsymbol{\theta}}(\boldsymbol{x},a)$ is an estimate of $f_{\boldsymbol{\theta}^*}(\boldsymbol{x},a)$. Following the conventional definition of confidence interval (Abbasi-Yadkori et al. (2011)), we define $\gamma$ and $U_{\boldsymbol{x},a}$ such that $|f_{\boldsymbol{\theta}^*}(\boldsymbol{x},a) - f_{\boldsymbol{\theta}}(\boldsymbol{x},a)| \leq \gamma U_{\boldsymbol{x},a}$ holds with probability at least 1-$\delta$, where $\gamma$ is a function of $\delta$ (typically the smaller $\delta$ is, the larger $\gamma$ is). Then $\gamma U_{\boldsymbol{x},a}$ measures the width of confidence interval of $f_{\boldsymbol{\theta}}(\boldsymbol{x},a)$ against its ground-truth $f_{\boldsymbol{\theta}^*}(\boldsymbol{x},a)$. This implies that $\beta^*(a|\boldsymbol{x}) \in \boldsymbol{B}_{\boldsymbol{x},a}$ with probability at least 1-$\delta$, where:

$$\boldsymbol{B}_{\boldsymbol{x},a} = \left[\frac{\hat{Z}\exp(-\gamma U_{\boldsymbol{x},a})}{Z^*}\hat{\beta}(a|\boldsymbol{x}), \frac{\hat{Z}\exp(\gamma U_{\boldsymbol{x},a})}{Z^*}\hat{\beta}(a|\boldsymbol{x})\right], \quad Z^* = \sum_{a'}\exp(f_{\boldsymbol{\theta}^*}(a'|\boldsymbol{x})), \hat{Z} = \sum_{a'}\exp(f_{\boldsymbol{\theta}}(a'|\boldsymbol{x})).$$

Since $\beta^*(a|\boldsymbol{x})$ can be any value in $\boldsymbol{B}_{\boldsymbol{x},a}$, we adopt the idea of robust optimization (Chen et al. (2020)) and find the optimal $\phi_{\boldsymbol{x},a}$ by solving the following optimization problem:

$$\min_{\phi_{\boldsymbol{x},a}} \max_{\beta_{\boldsymbol{x},a} \in \boldsymbol{B}_{\boldsymbol{x},a}} \quad \lambda \mathbb{E}_{\beta^*} \left[ \left( \frac{\beta_{\boldsymbol{x},a}}{\hat{\beta}(a|\boldsymbol{x})} \phi_{\boldsymbol{x},a} - 1 \right)^2 \right] + \mathbb{E}_{\beta^*} \left[ \frac{\pi_{\boldsymbol{\vartheta}}(a|\boldsymbol{x})^2}{\hat{\beta}(a|\boldsymbol{x})^2} \phi_{\boldsymbol{x},a}^2 \right]. \tag{8}$$

The following theorem derives a closed-form formula for the optimal solution of (8) .

**Theorem 2.** *Let $\eta_1, \eta_2 \in [\exp(-\gamma U_{\boldsymbol{x}}^{\max}), \exp(\gamma U_{\boldsymbol{x}}^{\max})]$, where $U_{\boldsymbol{x}}^{\max} = \max_a U_{\boldsymbol{x},a}$. The optimization problem in Eq.(8) has a closed-form solution as follows:*

$$\phi_{\boldsymbol{x},a}^* = \min \left( \lambda / \left[ \frac{\lambda}{\eta_1} \exp\left(-\gamma U_{\boldsymbol{x},a}\right) + \frac{\eta_1 \pi_{\boldsymbol{\vartheta}}(a|\boldsymbol{x})^2}{\hat{\beta}(a|\boldsymbol{x})^2 \exp(-\gamma U_{\boldsymbol{x},a})} \right], 2\eta_2 / \left[ \exp\left(\gamma U_{\boldsymbol{x},a}\right) + \exp\left(-\gamma U_{\boldsymbol{x},a}\right) \right] \right)$$

**Insights on $\phi_{\boldsymbol{x},a}^*$.** The second term of $\phi_{\boldsymbol{x},a}^*$ (i.e., $2\eta_2 / \left[\exp\left(\gamma U_{\boldsymbol{x},a}\right) + \exp\left(-\gamma U_{\boldsymbol{x},a}\right)\right]$ ) acts like a capping threshold to ensure $\phi_{\boldsymbol{x},a}^* \leq 2\eta_2$ holds even with small $\pi_{\boldsymbol{\vartheta}}(a|\boldsymbol{x})/\hat{\beta}(a|\boldsymbol{x})$ as shown in Lemma 1 in Appendix A.4. The key component is the first term, and Lemma 1 implies that:

- If the propensity score $\pi_{\boldsymbol{\vartheta}}(a|\boldsymbol{x})/\hat{\beta}(a|\boldsymbol{x})$ is above the threshold $\sqrt{\lambda}/\eta_1$, UIPS will assign a smaller weight to a sample with more inaccurate $\hat{\beta}(a|\boldsymbol{x})$ to prevent its distortion from a large propensity score but an inaccurate logging probability.
- If the propensity score $\pi_{\boldsymbol{\vartheta}}(a|\boldsymbol{x})/\hat{\beta}(a|\boldsymbol{x})$ is below the threshold but not small enough to activate the second term, then the propensity score at the worse case (i.e., taking $\boldsymbol{B}_{\boldsymbol{x},a}^- = \hat{\beta}(a|\boldsymbol{x})\hat{Z} \exp\left(-\gamma U_{\boldsymbol{x},a}\right)/Z^*$ as denominator) matters. If the propensity score at the worse case is under control, i.e., $\pi_{\boldsymbol{\vartheta}}(a|\boldsymbol{x})/\boldsymbol{B}_{\boldsymbol{x},a}^- < \sqrt{\lambda}$, a larger $U_{\boldsymbol{x},a}$ implies a small propensity score $\pi_{\boldsymbol{\vartheta}}(a|\boldsymbol{x})/\hat{\beta}(a|\boldsymbol{x})$, and UIPS tends to boost this safe sample with a higher $\phi_{\boldsymbol{x},a}^*$. Otherwise $\phi_{\boldsymbol{x},a}^*$ still decreases as $U_{\boldsymbol{x},a}$ becomes higher.

**Uncertainty estimation.** Now we describe how to calculate $U_{\boldsymbol{x},a}$, i.e., the uncertainty of the estimated $\hat{\beta}(a|\boldsymbol{x})$. In this work, we choose to estimate $\beta^*(a|\boldsymbol{x})$ using a neural network, due to its encouraging representation learning capacity. And various ways (Gal & Ghahramani (2016); Xu et al. (2021)) can be leveraged to perform the uncertainty estimation in a neural network. For example, (Gal & Ghahramani, 2016) proposed to estimate uncertainty using dropout; and (Xu et al., 2021) provided a theoretical bound. Here we adopt the result in (Xu et al. (2021)) due to its computational efficiency and theoretical soundness. Following the proof of Theorem 4.4 in (Xu et al. (2021)), given the logged dataset $D$, we can get with high probability $\exists \gamma$:

$$|f_{\boldsymbol{\theta}}(\boldsymbol{x}_n, a_n) - f_{\boldsymbol{\theta}^*}(\boldsymbol{x}_n, a_n))| \leq \gamma \sqrt{\boldsymbol{g}(\boldsymbol{x}_n, a_n)^T \boldsymbol{M}_D^{-1} \boldsymbol{g}(\boldsymbol{x}_n, a_n)}$$

where $\boldsymbol{g}(\boldsymbol{x}_n, a_n)$ is the gradient of $f_{\boldsymbol{\theta}}(\boldsymbol{x}_n, a_n)$ regarding to its last layer, i.e., $\boldsymbol{g}(\boldsymbol{x}_n, a_n) = \nabla_{\boldsymbol{\theta}_w} f_{\boldsymbol{\theta}}(\boldsymbol{x}_n, a_n)$, where $\boldsymbol{\theta}_w \subset \boldsymbol{\theta}$ is the parameter of the last layer of $f_{\boldsymbol{\theta}}(\boldsymbol{x}_n, a_n)$. And $\boldsymbol{M}_D = \sum_{n=1}^N \boldsymbol{g}(\boldsymbol{x}_n, a_n)\boldsymbol{g}(\boldsymbol{x}_n, a_n)^T$, implying $U_{\boldsymbol{x}_n, a_n} = \sqrt{\boldsymbol{g}(\boldsymbol{x}_n, a_n)^T \boldsymbol{M}_D^{-1} \boldsymbol{g}(\boldsymbol{x}_n, a_n)}$.

### 3.2 Policy Improvement

After getting the optimal $\phi_{\boldsymbol{x},a}^*$ as in Theorem 2, the policy $\pi_{\boldsymbol{\vartheta}}(a|\boldsymbol{x})$ can be updated by the following REINFORCE gradient:

$$\nabla_{\vartheta} V_{\text{UIPS}}(\pi_{\boldsymbol{\vartheta}}) = \mathbb{E}_{\beta^*} \left[ \frac{\pi_{\boldsymbol{\vartheta}}(a|\boldsymbol{x})}{\hat{\beta}(a|\boldsymbol{x})} \cdot \phi_{\boldsymbol{x},a}^* \cdot r_{\boldsymbol{x},a} \nabla_{\vartheta} \log(\pi_{\boldsymbol{\vartheta}}(a|\boldsymbol{x})) \right]. \tag{9}$$

UIPS then iterates policy evaluation and policy improvement for policy learning until converge. The whole algorithm framework and important notations are summarized in Algorithm 1 and Table 6 in Appendix A.1 respectively.

## 4 Empirical Evaluation

In this section, we evaluate UIPS on both synthetic datasets and three real-world datasets with unbiased data. We compare UIPS with the following baselines, which can be grouped into five categories:

- **Cross-Entropy (CE)**: A supervised learning method with the cross-entropy loss as its objective, which is the commonly used learning approach for a model with softmax output. No off-policy correction is performed in this method.

- **IPS-Cap** (Chen et al. (2019)): The standard IPS based off-policy learning, which prunes propensity scores to control variance, i.e., taking $\min(c, \frac{\pi_{\vartheta}(a|\boldsymbol{x})}{\hat{\beta}(a|\boldsymbol{x})})$ as the propensity score. Setting $c$ to a small value can reduce variance, but introduces bias.

- **MinVar** & **stableVar** (Zhan et al. (2021)), **Shrinkage** (Su et al. (2020)): This line of work improves off-policy evaluation estimators by reweighing each sample. For example, MinVar and stableVar reweigh each sample by $\frac{h_{\boldsymbol{x},a}}{\sum_{a'} h_{\boldsymbol{x},a'}}$ with $h_{\boldsymbol{x},a} = \frac{\hat{\beta}(a|\boldsymbol{x})}{\pi_{\vartheta}(a|\boldsymbol{x})^2}$ and $h_{\boldsymbol{x},a} = \frac{\sqrt{\hat{\beta}(a|\boldsymbol{x})}}{\pi_{\vartheta}(a|\boldsymbol{x})}$ respectively, since they find that $\pi_{\vartheta}(a|\boldsymbol{x})^2/\hat{\beta}(a|\boldsymbol{x})$ is directly related to variance. Su et al. (2020) proposes to shrink the propensity score by multiplying a weight $\lambda/(\lambda + \frac{\pi_{\vartheta}(a|\boldsymbol{x})^2}{\hat{\beta}(a|\boldsymbol{x})^2})$, which is a special case of the proposed UIPS with $U_{\boldsymbol{x},a} = 0$ and $\eta_1 = 1$. All these work simply treats $\hat{\beta}(a|\boldsymbol{x})$ as $\beta^*(a|\boldsymbol{x})$, and none of them consider the accuracy or uncertainty of $\hat{\beta}(a|\boldsymbol{x})$.

- **SNIPS** (Swaminathan & Joachims (2015c)), **BanditNet** (Joachims et al. (2018)), **POEM** (Swaminathan & Joachims (2015b)), **POXM** (Lopez et al. (2021)), **Adaptive** (Liu et al. (2022)): This line of work aims for more stable and accurate policy learning. For example, SNIPS normalizes the estimator by the sum of propensity scores in each batch. BanditNet extends SNIPS and leverages an additional Lagrangian term to normalize the estimator by an approximated sum of propensity scores of all samples. POEM jointly optimizes the estimator and its variance. POXM controls estimation variance by pruning samples with small logging probabilities. Adaptive proposes a new formulation to utilize negative samples.

- **UIPS-P** and **UIPS-O** : Two variants of our proposed UIPS with different ways of leveraging uncertainties. UIPS-P directly penalizes samples whose estimated logging probabilities are of high uncertainties, i.e., taking $\phi_{\boldsymbol{x},a} = 1.0/\exp(\gamma U_{\boldsymbol{x},a})$, which follows previous work on offline reinforcement learning (Wu et al. (2021); An et al. (2021)). UIPS-O adversarially uses the worst propensity scores $(\pi_{\vartheta}(a|\boldsymbol{x})/\boldsymbol{B}_{\boldsymbol{x},a}^-)$ for policy learning, i.e., $\phi_{\boldsymbol{x},a} = 1.0/\exp(-\gamma U_{\boldsymbol{x},a})$.

### 4.1 Synthetic Data

**Data generation.** Following previous work (Ma et al. (2020); Lopez et al. (2021)), we generate a synthetic dataset by a supervision-to-bandit conversion on Wiki10-31K dataset (Bhatia et al. (2016)), which is an extreme multi-label classification dataset. The Wiki10-31K dataset contains approximately 20K samples. Each sample is associated with a feature vector $\tilde{\boldsymbol{x}}$ of 101,938 dimensions and a label vector $\boldsymbol{y}_{\tilde{\boldsymbol{x}}}$ of 31K classes with more than one positive class. Let $\boldsymbol{y}_{\tilde{\boldsymbol{x}},a}$ denote the label of class $a$ under $\tilde{\boldsymbol{x}}$ and we take each class as an action. We adopt the Wiki10-31K dataset rather than ones in the UCI machine learning repository (Swaminathan & Joachims (2015a)), since it will be much harder with such a large action space.

We then split the dataset into train, validation, test sets with size 11K:3K:6K. The test set is from the official split. Since the original feature vector $\tilde{\boldsymbol{x}}$ is too sparse, for ease of learning, we first embed it to dimension $d$ by $\boldsymbol{x} = \boldsymbol{W}\tilde{\boldsymbol{x}}$, and synthesize the ground-truth logging policy $\beta^*(a|\boldsymbol{x})$ by:

$$\beta^*(a|\boldsymbol{x}) = \frac{\exp(\boldsymbol{x}^T \boldsymbol{\theta}_a^*/\tau)}{\sum_{a'} \exp(\boldsymbol{x}^T \boldsymbol{\theta}_{a'}^*/\tau)}, \tag{10}$$

where $\boldsymbol{W}$ and $\{\boldsymbol{\theta}_a^*\}$ are pre-learned parameters by applying a logistic regression model on the train set, $\tau$ is a hyper-parameter that controls the skewness of logging distribution. A small value of $\tau$ leads to a near-deterministic policy, while a larger $\tau$ makes logging policy smoother. Due to space limit, more details on data generation and implementation can be found in Appendix A.2.

**Evaluation metrics**. To evaluate the learned policy $\pi_{\vartheta}(a|\boldsymbol{x})$, we calculate Precision@K (P@K), Recall@K (R@K) and NDCG@K as in previous work (Lopez et al. (2021); Ma et al. (2020)). Higher P@K, R@K and NDCG@K imply a better policy.

Table 1 shows the mean performance and standard deviations of all algorithms under 10 random seeds on three synthetic datasets generated under different $\tau$. Since the ground-truth logging policy is accessible on the synthetic datasets, we include a new baseline IPS-GT, which depicts the performance the IPS estimator can achieve, assuming the ground-truth logging probabilities are known and sample size is sufficiently large. We calculate p-value under t-test between UIPS and the best baseline on each dataset to investigate the significance of improvement. First, we can observe that UIPS achieves similar and even better performance than IPS-GT when $\tau = 0.5$ and $\tau = 1$, but

| Algorithm | $\tau = 0.5$ | | | $\tau = 1$ | | | $\tau = 2$ | | |
|---|---|---|---|---|---|---|---|---|---|
| | P@5 | R@5 | NDCG@5 | P@5 | R@5 | NDCG@5 | P@5 | R@5 | NDCG@5 |
| IPS-GT | $0.5589_{\pm 1e^{-3}}$ | $0.1582_{\pm 6e^{-4}}$ | $0.6093_{\pm 1e^{-3}}$ | $0.5526_{\pm 2e^{-3}}$ | $0.1565_{\pm 6e^{-4}}$ | $0.6007_{\pm 1e^{-3}}$ | $0.5531_{\pm 2e^{-3}}$ | $0.1557_{\pm 7e^{-4}}$ | $0.6037_{\pm 1e^{-3}}$ |
| CE | $\underline{0.5553_{\pm 6e^{-4}}}$ | $\underline{0.1573_{\pm 2e^{-4}}}$ | $\underline{0.6037_{\pm 5e^{-4}}}$ | $0.5510_{\pm 6e^{-4}}$ | $0.1561_{\pm 2e^{-4}}$ | $0.5995_{\pm 4e^{-4}}$ | $0.5386_{\pm 2e^{-3}}$ | $0.1524_{\pm 7e^{-4}}$ | $0.5874_{\pm 2e^{-3}}$ |
| IPS-Cap | $0.5515_{\pm 2e^{-3}}$ | $0.1553_{\pm 8e^{-4}}$ | $0.6031_{\pm 2e^{-3}}$ | $0.5526_{\pm 2e^{-3}}$ | $0.1561_{\pm 6e^{-4}}$ | $0.6016_{\pm 1e^{-3}}$ | $\underline{0.5409_{\pm 3e^{-3}}}$ | $\underline{0.1529_{\pm 9e^{-4}}}$ | $\underline{0.5901_{\pm 2e^{-3}}}$ |
| MinVar | $0.5340_{\pm 2e^{-3}}$ | $0.1509_{\pm 6e^{-4}}$ | $0.5857_{\pm 2e^{-3}}$ | $0.5282_{\pm 2e^{-3}}$ | $0.1491_{\pm 7e^{-4}}$ | $0.5791_{\pm 2e^{-3}}$ | $0.5036_{\pm 4e^{-3}}$ | $0.1415_{\pm 1e^{-3}}$ | $0.5543_{\pm 3e^{-3}}$ |
| StableVar | $0.4577_{\pm 5e^{-3}}$ | $0.1310_{\pm 1e^{-3}}$ | $0.5111_{\pm 2e^{-3}}$ | $0.5373_{\pm 3e^{-3}}$ | $0.1523_{\pm 9e^{-4}}$ | $0.5866_{\pm 3e^{-3}}$ | $0.5279_{\pm 3e^{-3}}$ | $0.1492_{\pm 8e^{-4}}$ | $0.5781_{\pm 3e^{-3}}$ |
| Shrinkage | $0.5526_{\pm 2e^{-3}}$ | $0.1562_{\pm 7e^{-4}}$ | $0.6024_{\pm 1e^{-3}}$ | $0.5499_{\pm 4e^{-3}}$ | $0.1545_{\pm 1e^{-3}}$ | $\underline{0.6040_{\pm 3e^{-3}}}$ | $0.5347_{\pm 2e^{-3}}$ | $0.1513_{\pm 6e^{-4}}$ | $0.5824_{\pm 2e^{-3}}$ |
| SNIPS | $0.2616_{\pm 6e^{-2}}$ | $0.0749_{\pm 2e^{-2}}$ | $0.3150_{\pm 7e^{-2}}$ | $0.3538_{\pm 5e^{-2}}$ | $0.0987_{\pm 1e^{-2}}$ | $0.4144_{\pm 6e^{-2}}$ | $0.4379_{\pm 3e^{-2}}$ | $0.1226_{\pm 9e^{-3}}$ | $0.5177_{\pm 3e^{-2}}$ |
| BanditNet | $0.4011_{\pm 3e^{-2}}$ | $0.1131_{\pm 8e^{-3}}$ | $0.4830_{\pm 2e^{-2}}$ | $0.3894_{\pm 4e^{-2}}$ | $0.1095_{\pm 1e^{-2}}$ | $0.4741_{\pm 3e^{-2}}$ | $0.4122_{\pm 3e^{-2}}$ | $0.1153_{\pm 8e^{-3}}$ | $0.4934_{\pm 3e^{-2}}$ |
| POEM | $0.5480_{\pm 2e^{-3}}$ | $0.1539_{\pm 8e^{-4}}$ | $0.6008_{\pm 2e^{-3}}$ | $0.5502_{\pm 2e^{-3}}$ | $0.1551_{\pm 6e^{-4}}$ | $0.6000_{\pm 2e^{-3}}$ | $0.5399_{\pm 2e^{-3}}$ | $0.1526_{\pm 8e^{-4}}$ | $0.5893_{\pm 2e^{-3}}$ |
| POXM | $0.4006_{\pm 3e^{-2}}$ | $0.1130_{\pm 8e^{-3}}$ | $0.4828_{\pm 2e^{-2}}$ | $0.3616_{\pm 4e^{-2}}$ | $0.1019_{\pm 1e^{-2}}$ | $0.4522_{\pm 4e^{-2}}$ | $0.3816_{\pm 4e^{-2}}$ | $0.1069_{\pm 1e^{-2}}$ | $0.4680_{\pm 4e^{-2}}$ |
| Adaptive | $0.3831_{\pm 2e^{-2}}$ | $0.1050_{\pm 4e^{-3}}$ | $0.4382_{\pm 2e^{-2}}$ | $0.4734_{\pm 4e^{-3}}$ | $0.1325_{\pm 1e^{-3}}$ | $0.5326_{\pm 3e^{-3}}$ | $0.3936_{\pm 1e^{-2}}$ | $0.1097_{\pm 4e^{-3}}$ | $0.4368_{\pm 2e^{-2}}$ |
| UIPS-P | $0.4019_{\pm 3e^{-2}}$ | $0.1131_{\pm 1e^{-2}}$ | $0.4831_{\pm 3e^{-2}}$ | $0.3904_{\pm 4e^{-2}}$ | $0.1096_{\pm 1e^{-2}}$ | $0.4749_{\pm 3e^{-2}}$ | $0.4109_{\pm 3e^{-2}}$ | $0.1149_{\pm 1e^{-2}}$ | $0.4922_{\pm 3e^{-2}}$ |
| UIPS-O | $0.4135_{\pm 4e^{-2}}$ | $0.1167_{\pm 1e^{-2}}$ | $0.4954_{\pm 4e^{-2}}$ | $0.3896_{\pm 4e^{-2}}$ | $0.1096_{\pm 1e^{-2}}$ | $0.4739_{\pm 3e^{-2}}$ | $0.4519_{\pm 3e^{-2}}$ | $0.1268_{\pm 8e^{-3}}$ | $0.5296_{\pm 2e^{-2}}$ |
| UIPS | $\mathbf{0.5608_{\pm 2e^{-3}}}$ | $\mathbf{0.1589_{\pm 8e^{-4}}}$ | $\mathbf{0.6113_{\pm 3e^{-3}}}$ | $\mathbf{0.5572_{\pm 2e^{-3}}}$ | $\mathbf{0.1571_{\pm 8e^{-4}}}$ | $\mathbf{0.6074_{\pm 2e^{-3}}}$ | $\mathbf{0.5432_{\pm 3e^{-3}}}$ | $\mathbf{0.1534_{\pm 8e^{-4}}}$ | $\mathbf{0.5946_{\pm 2e^{-3}}}$ |
| p-value | $4e^{-6}$ | $4e^{-5}$ | $2e^{-10}$ | $2e^{-7}$ | $2e^{-3}$ | $4e^{-10}$ | $1e^{-1}$ | $2e^{-1}$ | $4e^{-2}$ |

Table 1: Experimental results on synthetic datasets. The best and second best results are highlighted with **bold** and underline respectively. The p-value under the t-test between UIPS and the best baseline on each dataset is also provided.

| Algorithm | Low Frequent Action Related Samples ( High Uncertainty) | | | High Frequent Action Related Samples (Low Uncertainty) | | |
|---|---|---|---|---|---|---|
| | P@5(RI) | R@5(RI) | NDCG@5(RI) | P@5(RI) | R@5(RI) | NDCG@5(RI) |
| CE | 0.5186 | 0.1228 | 0.5521 | 0.5931 | 0.1921 | 0.6575 |
| IPS-Cap | 0.5170(-0.31%) | 0.1218(-0.81%) | 0.5539(+0.32%) | 0.5996(+1.10%) | 0.1935(+0.73%) | 0.6647(+1.10%) |
| Shrinkage | 0.5145(-0.79%) | 0.1212(-1.30%) | 0.5519(-0.04%) | 0.5982(+0.86%) | 0.1931(+0.52%) | 0.6628(+0.81%) |
| UIPS | 0.5276(+1.74%) | 0.1250(+1.79%) | 0.5623(+1.85%) | 0.6055(+2.09%) | 0.1961(+2.08%) | 0.6715(+2.13%) |

Table 2: Performance under different uncertainties.

| Algorithm | MSE |
|---|---|
| IPS-CaP | 0.4953 |
| minVar | 0.8928 |
| stableVar | 0.8112 |
| Shrinage | 0.5125 |
| UIPS | 0.4516 |

Table 4: MSE of different off-policy evaluation estimators.

Table 3: Effect of $\lambda$ and $\gamma$ on NDCG@5.

performs worse than IPS-GT on the dataset with $\tau = 2$. Although IPS-GT can access the ground-truth logging probabilities, it still suffers from high variance caused by samples with small logging probabilities, which is the main cause of its worse performance when $\tau = 0.5$ and $\tau = 1$. When the ground-truth logging policy is smoother (e.g., $\tau = 2$), the variance of the IPS estimator becomes much smaller, and off-policy correction with the ground-truth logging probabilities, rather than the estimated ones, leads to better model performance. We can then observe that as $\tau$ increases, i.e., the probability of selecting positive actions decreases, the performance of most algorithms drop, including CE, IPS-Cap, UIPS, Shrinkage, POEM, Adaptive, etc. However, UIPS still achieves the best performance on all three datasets under all three metrics. And as $\tau$ decreases, the improvement of UIPS becomes larger and more significant. SNIPS, BanditNet , POXM are more robust to small logging probabilities of positive actions. UIPS consistently outperforms Shrinkage (a special case of UIPS with uncertainties always being zero) on all three datasets, demonstrating the benefits of considering the estimation uncertainty. Finally, regardless of the scale of propensity scores, blindly reweighing through uncertainties also leads to poor performance, as shown by UIPS-P and UIPS-O.

**Performance under different uncertainty levels.** As shown in Figure 1, low-frequency actions in the logged dataset suffer higher uncertainties in their propensity estimation. Thus, we divide the test set into two subsets according to the average frequency of associated actions, where the uncertainty in the subset associated with low-frequency actions is on average 9% higher than that in the subset associated with high-frequency actions. Table 2 shows the results on these two subsets when $\tau = 0.5$. We only report the results of the best three baselines due to space limit. One can clearly observe that only UIPS performed better than CE on the test set associated with low-frequency actions, implying the advantage of UIPS in dealing with the inaccurately estimated logging probabilities.

| Algorithm | Yahoo | | | Coat | | | KuaiRec | | |
|---|---|---|---|---|---|---|---|---|---|
| | P@5 | R@5 | NDCG@5 | P@5 | R@5 | NDCG@5 | P@50 | R@50 | NDCG@50 |
| CE | $0.2819_{\pm 2e^{-3}}$ | $0.7594_{\pm 6e^{-3}}$ | $0.6073_{\pm 7e^{-3}}$ | $0.2799_{\pm 5e^{-3}}$ | $0.4618_{\pm 1e^{-2}}$ | $0.4529_{\pm 7e^{-3}}$ | $0.8802_{\pm 2e^{-3}}$ | $0.0240_{\pm 8e^{-5}}$ | $0.8810_{\pm 6e^{-3}}$ |
| IPS-Cap | $0.2751_{\pm 2e^{-3}}$ | $0.7419_{\pm 8e^{-3}}$ | $0.5928_{\pm 7e^{-3}}$ | $0.2758_{\pm 6e^{-3}}$ | $0.4582_{\pm 7e^{-3}}$ | $0.4399_{\pm 9e^{-3}}$ | $0.8750_{\pm 3e^{-3}}$ | $0.0238_{\pm 7e^{-5}}$ | $0.8788_{\pm 5e^{-3}}$ |
| MinVar | $0.2843_{\pm 4e^{-3}}$ | $\underline{0.7685_{\pm 1e^{-2}}}$ | $0.6168_{\pm 1e^{-2}}$ | $0.2813_{\pm 3e^{-3}}$ | $\underline{0.4668_{\pm 9e^{-3}}}$ | $0.4414_{\pm 8e^{-3}}$ | $0.8827_{\pm 1e^{-3}}$ | $0.0240_{\pm 5e^{-5}}$ | $0.8886_{\pm 2e^{-3}}$ |
| StableVar | $0.2787_{\pm 2e^{-3}}$ | $0.7499_{\pm 7e^{-3}}$ | $0.5919_{\pm 7e^{-3}}$ | $\underline{0.2840_{\pm 3e^{-3}}}$ | $0.4662_{\pm 5e^{-3}}$ | $0.4393_{\pm 7e^{-3}}$ | $0.8524_{\pm 7e^{-3}}$ | $0.0231_{\pm 2e^{-4}}$ | $0.8570_{\pm 4e^{-3}}$ |
| Shrinkage | $0.2843_{\pm 3e^{-3}}$ | $0.7654_{\pm 8e^{-3}}$ | $\underline{0.6204_{\pm 7e^{-3}}}$ | $0.2790_{\pm 5e^{-3}}$ | $0.4636_{\pm 4e^{-3}}$ | $\underline{0.4464_{\pm 1e^{-2}}}$ | $0.8744_{\pm 3e^{-3}}$ | $0.0238_{\pm 9e^{-5}}$ | $0.8771_{\pm 6e^{-3}}$ |
| SNIPS | $0.2222_{\pm 4e^{-3}}$ | $0.5828_{\pm 1e^{-2}}$ | $0.4357_{\pm 1e^{-2}}$ | $0.2643_{\pm 7e^{-3}}$ | $0.4287_{\pm 1e^{-2}}$ | $0.4009_{\pm 9e^{-3}}$ | $0.8411_{\pm 6e^{-3}}$ | $0.0228_{\pm 2e^{-4}}$ | $0.8431_{\pm 6e^{-3}}$ |
| BanditNet | $0.2413_{\pm 8e^{-3}}$ | $0.6442_{\pm 2e^{-2}}$ | $0.4988_{\pm 2e^{-2}}$ | $0.2781_{\pm 8e^{-3}}$ | $0.4527_{\pm 1e^{-2}}$ | $0.4251_{\pm 1e^{-2}}$ | $0.8758_{\pm 5e^{-3}}$ | $0.0239_{\pm 2e^{-4}}$ | $0.8810_{\pm 4e^{-3}}$ |
| POEM | $0.2732_{\pm 3e^{-3}}$ | $0.7357_{\pm 1e^{-2}}$ | $0.5880_{\pm 1e^{-2}}$ | $0.2791_{\pm 4e^{-3}}$ | $0.4566_{\pm 6e^{-3}}$ | $0.4375_{\pm 6e^{-3}}$ | $0.7785_{\pm 1e^{-2}}$ | $0.0210_{\pm 2e^{-4}}$ | $0.7779_{\pm 6e^{-3}}$ |
| POXM | $0.2250_{\pm 5e^{-3}}$ | $0.5940_{\pm 1e^{-2}}$ | $0.4542_{\pm 2e^{-2}}$ | $0.2663_{\pm 6e^{-3}}$ | $0.4308_{\pm 9e^{-3}}$ | $0.4006_{\pm 1e^{-2}}$ | $\underline{0.8962_{\pm 1e^{-2}}}$ | $\underline{0.0245_{\pm 4e^{-4}}}$ | $\underline{0.9041_{\pm 1e^{-2}}}$ |
| Adaptive | $0.2762_{\pm 3e^{-3}}$ | $0.7451_{\pm 9e^{-3}}$ | $0.5919_{\pm 8e^{-3}}$ | $0.2830_{\pm 3e^{-3}}$ | $0.4634_{\pm 5e^{-3}}$ | $0.4217_{\pm 5e^{-3}}$ | $0.8375_{\pm 1e^{-2}}$ | $0.0227_{\pm 4e^{-4}}$ | $0.8460_{\pm 1e^{-2}}$ |
| UIPS-P | $0.1829_{\pm 8e^{-3}}$ | $0.4560_{\pm 3e^{-2}}$ | $0.3300_{\pm 1e^{-2}}$ | $0.2685_{\pm 7e^{-3}}$ | $0.4364_{\pm 9e^{-3}}$ | $0.4087_{\pm 7e^{-3}}$ | $0.8638_{\pm 8e^{-3}}$ | $0.0235_{\pm 3e^{-4}}$ | $0.8685_{\pm 7e^{-3}}$ |
| UIPS-O | $0.1947_{\pm 3e^{-3}}$ | $0.4959_{\pm 1e^{-2}}$ | $0.3600_{\pm 8e^{-3}}$ | $0.2657_{\pm 5e^{-3}}$ | $0.4306_{\pm 9e^{-3}}$ | $0.4146_{\pm 9e^{-3}}$ | $0.8651_{\pm 8e^{-3}}$ | $0.0235_{\pm 2e^{-4}}$ | $0.8697_{\pm 7e^{-3}}$ |
| UIPS | $\mathbf{0.2868_{\pm 2e^{-3}}}$ | $\mathbf{0.7742_{\pm 5e^{-3}}}$ | $\mathbf{0.6274_{\pm 5e^{-3}}}$ | $\mathbf{0.2877_{\pm 3e^{-3}}}$ | $\mathbf{0.4757_{\pm 5e^{-3}}}$ | $\mathbf{0.4576_{\pm 8e^{-3}}}$ | $\mathbf{0.9120_{\pm 1e^{-3}}}$ | $\mathbf{0.0250_{\pm 5e^{-5}}}$ | $\mathbf{0.9174_{\pm 7e^{-4}}}$ |
| P-value | $4e^{-2}$ | $1e^{-2}$ | $3e^{-2}$ | $2e^{-2}$ | $6e^{-4}$ | $5e^{-5}$ | $6e^{-4}$ | $6e^{-4}$ | $1e^{-3}$ |

Table 5: Experimental results on real-world datasets. The best and second best results are highlighted with **bold** and underline respectively. The p-value under the t-test between UIPS and the best baseline on each dataset is also provided.

**Ablation Study.** In this experiment, we aim to answer two questions: (1) Can $\hat{V}_{\mathrm{UIPS}}(\pi_{\boldsymbol{\vartheta}})$ in Eq. (5) lead to more accurate off-policy evaluation? (2) How will UIPS perform with different hyperparameters. Due to space limit, we report results on synthetic dataset with $\tau = 0.5$.

To answer the first question, we evaluate the following $\epsilon$-greedy policy: $\pi(a|\boldsymbol{x}) = \frac{1-\epsilon}{|M_x|} \cdot \mathbb{I}\{a \in M_x\} + \epsilon/|\mathcal{A}|$, where $M_x$ contains all positive actions associated with feature vector $\boldsymbol{x}$. Then for each $\boldsymbol{x}$ in the test set, we sample 1K data points in a similar way as discussed previously to calculate the value of estimators. Table 4 shows the MSE of the estimators to ground-truth policy value under 20 different random seeds. We only compared with baselines on off-policy evaluation estimator, i.e., IPS-Cap, MinVar, StableVar and Shrinkage. One can observe from Table 4 that UIPS does lead to the smallest MSE, implying the most accurate off-policy evaluation.

For the second question, $\gamma$ and $\eta_1^2/\lambda$ are the two most important hyperparameters as discussed in Appendix A.2.1. Thus we fix $\eta_1, \eta_2$, and vary $\lambda$ and $\gamma$ to track the performance of UIPS. Recall that a larger $\gamma$ implies a higher chance the derived interval contains $\beta^*(a|\boldsymbol{x})$, while $\sqrt{\lambda}/\eta_1$ is closely related to how UIPS works as discussed in "Insights on $\phi^*_{\boldsymbol{x},a}$" in Section 3.1. Figure 3 reports NDCG@5 under diferent $\gamma$ and $\lambda$. Results on P@5 and R@5 can be found in Appendix A.2.1. We can observe that to make UIPS perform, $\boldsymbol{B}_{\boldsymbol{x},a}$ needs to be of high confidence, e.g., $\gamma = 25$ performed the best when $\tau = 0.5$. Moreover, the threshold $\sqrt{\lambda}/\eta_1$ cannot be too small or too large.

## 4.2 Real-World Data

Off-policy learning has its utility in recommendation scenarios (Chen et al. (2019); Ma et al. (2020)), where context vector $\boldsymbol{x}$ denotes the state of a user and each candidate item is taken as an action. To further demonstrate the efficiency of UIPS in real-world scenarios, we evaluate it on three recommendation datasets with unbiased testing data: (1) Yahoo!R3[1]; (2)Coat[2]; (3)KuaiRec (Gao et al. (2022)), from music, fashion and micro-video recommendation scenario respectively. All these datasets contain an unbiased test set collected from a randomized controlled trial where items are randomly selected. The statistics of the three datasets and implementation details, e.g., model architectures and dataset splits, can be found in Appendix A.2.2.

We still adopt P@K, R@K and NDCG@K as our evaluation metrics. Following (Ding et al. (2022)), we take $K = 5$ on Yahoo!R3 and Coat datasets, and $K = 50$ on KuaiRec dataset. The p-value under the t-test between UIPS and the best baseline on each dataset is also reported to investigate the significance of the improvements. We can first observe that on all three datasets, the proposed UIPS achieves the highest precision, recall and NDCG. IPS-Cap cannot outperform CE due to the inaccuracy of the estimated logging probabilities. BanditNet, POEM and POXM tend to perform better with a larger action space, while MinVar, StableVar and Shrinkage as well as Adaptive are more suitable for scenarios with small action size. UIPS still outperforms Shrinkage, highlighting the importance of modeling uncertainty in the estimated logging policy. However, reweighing based solely on uncertainties, ignoring the corresponding propensity scores, will also lead to poor performance, as shown by UIPS-P and UIPS-O.

---

[1] https://webscope.sandbox.yahoo.com/
[2] https://www.cs.cornell.edu/~schnabts/mnar/

## 5 RELATED WORK

This work is the first of its kind to take into consideration the uncertainty of the estimated logging policy for improved policy learning. The following two lines of work are related to this paper.

**Off-policy learning.** In many real-world applications, such as search engines, recommender systems, etc., interactive online model update is expensive and risky (Jiang & Li (2016)). Off-policy learning has therefore attracted increasing interest, since it can leverage the already logged feedback data (Agarwal et al. (2019); Chen et al. (2019); Liu et al. (2022)). The main challenge in off-policy learning is how to address the mismatch between the logging policy and the learning policy. One line of work (Achiam et al. (2017); Schulman et al. (2015)) circumvents this by constraining the learning policy not too far from the logging policy. However, such constraint is too restrictive thus not applicable in some scenarios such as recommender systems where user behaviors and items change rapidly. Another more common and widely-applied approach is to leverage Inverse Propensity Score (IPS) method to correct the discrepancy between two policies. And various methods are proposed for stabilized learning (Swaminathan & Joachims (2015c;a;b)) and variance control (Lopez et al. (2021); Liu et al. (2022)) on top of IPS. However, all these work directly use the estimated logging policy for off-policy correction, leading to sub-optimal performance as shown in our experiments. Some other work further extend IPS-based off-policy learning for more complex problems, such as slate recommendation (Swaminathan et al. (2017)), two-stage recommender systems (Ma et al. (2020)), etc. But they still fail to realize the effect of accuracy of the estimated logging policy. A recent work (Ding et al. (2022)) on causal recommendation also argues that propensity scores may not be correct due to unobserved confounders. However, they assume the effect of unobserved confounder for any sample can be bounded by a pre-defined hyper-parameter, and adversarially search for the worst-case propensity to update model parameters. Adapting to off-policy learning, it is a special case of our UIPS-O variant with uncertainty as a pre-defined constant.

Off-policy learning can be directly built on off-policy evaluation. In this line of research, several work (Su et al. (2020); Zhan et al. (2021)) also propose to control the high variance of learning caused by small logging probabilities by instance reweighing. However, they directly take the estimated logging policy as true logging policy for correction, thus worse than UIPS as shown in experiments. A recent work (Saito & Joachims (2022)) assumes additional structure in action space and proposes the marginalized IPS. Instead, our work considers the uncertainty when estimating the logging policy and thus does not add new assumptions about the problem space.

**Uncertainty-aware Learning.** Estimation uncertainty has been extensively used for making trade-offs between exploration and exploitation in online learning (Xu et al. (2021); Zhou et al. (2020); Abbasi-Yadkori et al. (2011)). Recently, several work on offline reinforcement learning (Wu et al. (2021); An et al. (2021); Bai et al. (2022)) penalize the value function of out-of-distribution states and actions by directly subtracting uncertainty to tackle the extrapolating error. However, blindly penalizing samples of high uncertainty (i.e., UIPS-P) is problematic, as shown in our experiments. Proper correction depends on both uncertainty in logging policy estimation and the actual value of estimated logging probabilities.

## 6 CONCLUSION

In this paper, we propose a novel Uncertainty-aware Inverse Propensity Score estimator (UIPS) to explicitly model the uncertainty about the estimated logging policy for improved off-policy learning. UIPS weighs each logged instance to approach the ground-truth estimator and a closed-form solution of the optimal weight is derived by minimizing the upper bound of the mean squared error (MSE). An improved policy can be obtained by optimizing the resulting estimator. Extensive experiments on synthetic datasets and three real-world datasets demonstrate the efficiency of UIPS .

As demonstrated in this work, explicitly modeling the uncertainty of the estimated logging policy is crucial for effective off-policy learning; but the best use of this uncertainty is not to simply down-weigh or drop instances with uncertain estimations, but to balance it with the actually estimated logging probabilities in a per-instance basis. As our future work, it is promising to investigate how UIPS can be extended to value-based learning methods, e.g., actor-critics. And on the other hand, it is also important to analyze how tight our upper bound analysis of MSE is; and if possible, find new ways to tighten it for improvements.

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

# A  APPENDIX

## A.1  NOTATIONS AND ALGORITHM FRAMEWORK.

For ease of reading, we list important notations in Table 6 and summarize the main framework of the proposed UIPS in Algorithm 1.

| Notation | Description |
|---|---|
| $\mathcal{X}$ | context space |
| $\mathcal{A}$ | action set |
| $\boldsymbol{x} \in R^d$ | context vector |
| $a$ | action |
| $r_{\boldsymbol{x},a}$ | reward |
| $\pi(a\|\boldsymbol{x})$ | targeted policy to evaluate |
| $\beta^*(a\|\boldsymbol{x})$ | the unknown ground-truth logging policy |
| $\hat{\beta}(a\|\boldsymbol{x})$ | the estimated logging policy |
| $V(\pi)$ | value function |
| $D := \{(\boldsymbol{x}_n, a_n, r_{\boldsymbol{x}_n,a_n}) \| n \in [N]\}$ | logged dataset containing $N$ samples |
| $\phi^*_{\boldsymbol{x},a}$ | the optimal uncertainty-aware weight |
| $f_{\boldsymbol{\theta}^*}(\boldsymbol{x}, a)$ | the unknown ground-truth function that generates $\beta^*(a\|\boldsymbol{x}) = \frac{\exp(f_{\boldsymbol{\theta}*}(\boldsymbol{x},a))}{\sum_{a'} \exp(f_{\boldsymbol{\theta}*}(\boldsymbol{x},a'))}$ |
| $f_{\boldsymbol{\theta}}(\boldsymbol{x}, a)$ | the estimate of $f_{\boldsymbol{\theta}^*}(\boldsymbol{x}, a)$ that generates $\hat{\beta}(a\|\boldsymbol{x})$ |
| $\boldsymbol{B}_{\boldsymbol{x},a}$ | confidence interval of $\hat{\beta}(a\|\boldsymbol{x})$ |
| $U_{\boldsymbol{x},a}$ | uncertainty defined as $\|f_{\boldsymbol{\theta}*}(\boldsymbol{x}, a) - f_{\boldsymbol{\theta}}(\boldsymbol{x}, a)\| \leq \gamma U_{\boldsymbol{x},a}$ |
| $\boldsymbol{g}(\boldsymbol{x}_n, a_n)$ | gradient of $f_{\boldsymbol{\theta}}(\boldsymbol{x}, a)$ regarding to the last layer. |

Table 6: Notations

**Computation Cost.** The additional computation cost of UIPS over IPS comes from two parts:

- Pre-calculating uncertainties (line 1-5 in Algorithm 1) : This part calculates uncertainty of the logging probability for each $(s, a)$ pair, and "it only needs to be executed once". The computational cost of this step is $O(Nd^2 + d^3)$, where $O(Nd^2)$ is for calculating uncertainties in each $(s, a)$ pair and $O(d^3)$ is for matrix inverse.
- Calculating $\phi^*_{\boldsymbol{x},a}$ during training (line 8 in Algorithm 1): It only takes O(1) time, the same computational cost as calculating IPS score.

Note that calculating logging probability for each sample, which is essential for both UIPS and IPS, takes $O(Nd|\mathcal{A}|)$ time. Since the dimension $d$ is usually much less than action size $|\mathcal{A}|$ and samples size $N$, UIPS does not introduce significant computational overhead compared to the original IPS solution.

## A.2  EXPERIMENTS DETAILS

### A.2.1  SYNTHETIC DATA

**Data generation.** Given the ground-truth logging policy $\beta^*(a|\boldsymbol{x})$, we generate the logged dataset as follows. For each sample in train set, we first get the embedded context vector $\boldsymbol{x}$ form its original

---

**Algorithm 1: UIPS**

---

**Input:** *The logged dataset* $D := \{(\boldsymbol{x}_n, a_n, r_{\boldsymbol{x}_n, a_n}) | n \in [N]\}$, *the estimated logging policy*
      *model* $\hat{\beta}(a|\boldsymbol{x}) = \frac{\exp(f_{\boldsymbol{\theta}}(\boldsymbol{x},a))}{\sum_{a'} \exp(f_{\boldsymbol{\theta}}(\boldsymbol{x},a'))}$, *latent dimension d.*

**Init:** $\boldsymbol{M}_D = \boldsymbol{I}_{d \times d}$
`// calculate` $\boldsymbol{M}_D$ `for uncertainty calculation.`

1   **for** $n = 1, 2, ..., N$ **do**
2      $\boldsymbol{M}_D = \boldsymbol{M}_D + \nabla_{\boldsymbol{\theta}} f_{\boldsymbol{\theta}}(\boldsymbol{x}_n, a_n) \nabla_{\boldsymbol{\theta}} f_{\boldsymbol{\theta}}(\boldsymbol{x}_n, a_n)^T$ ;
3   $\boldsymbol{M}_D^{\text{inv}} = \text{inv}(\boldsymbol{M}_D)$ ;
4   **for** $n = 1, 2, ..., N$ **do**
5      $U_{\boldsymbol{x}_n, a_n} = \sqrt{\nabla_{\boldsymbol{\theta}} f_{\boldsymbol{\theta}}(\boldsymbol{x}_n, a_n)^T \boldsymbol{M}_D^{\text{inv}} \nabla_{\boldsymbol{\theta}} f_{\boldsymbol{\theta}}(\boldsymbol{x}_n, a_n)}$;
   `// Main part of UIPS`
6   **while** *not converge* **do**
7      **for** $n = 1, 2, ..., N$ **do**
8          *Calculating* $\phi_{\boldsymbol{x}_n, a_n}^*$ *as in Theorem 2* ;
9          *Calculating gradients as in Equation (9) and updating* $\pi_{\boldsymbol{\vartheta}}(a|\boldsymbol{x})$.

**Output:** *The learnt policy* $\pi_{\boldsymbol{\vartheta}}(a|\boldsymbol{x})$.

---

feature vector $\tilde{\boldsymbol{x}}$. We then sample an action $a$ according to $\beta^*(a|\boldsymbol{x})$, and obtain the reward $r_{\boldsymbol{x},a} = \boldsymbol{y}_{\tilde{\boldsymbol{x}},a}$, resulting a bandit feedback $(\boldsymbol{x}, a, r_{\boldsymbol{x},a})$, where $\boldsymbol{y}_{\tilde{\boldsymbol{x}},a}$ is the label of class $a$ under the original feature vector $\tilde{\boldsymbol{x}}$. We repeat above process $N$ times to collect the logged dataset. In our experiments, we take $d = 64, N = 100$.

**Implementation Details.** We model the logging policy as in Equation (7) with $f_{\boldsymbol{\theta}}(\boldsymbol{x}, a) = \boldsymbol{x}^T \boldsymbol{\theta}_a$, where $\{\boldsymbol{\theta}_a\}$ are parameters to learn. To train the logging policy, we take all samples in the logged dataset $D$ as positive instances, and randomly sample non-selected actions as negative instances as in (Chen et al. (2019)). We use grid search to select the hyperparameters based on the model's performance on validation dataset: the learning rate was searched in $\{1e^{-5}, 1e^{-4}, 1e^{-3}, 1e^{-2}\}$; $\lambda, \gamma, \eta_1$ were searched in $\{0.5, 0.1, 1, 2, 5, 10, 15, 20, 25, 30, 40, 50\}$. And $\eta_2$ was searched in $\{1, 10, 100, 1000\}$. For baseline algorithms, we perform a similar grid search as mentioned above, and the search range follows the original papers.

**Ablation Study: Hyperparameter tuning.** Although UIPS has four hyperparameters ($\lambda, \gamma, \eta_1$, and $\eta_2$), one only needs to carefully finetune two of them, i.e., $\gamma$ and $\eta_1^2/\lambda$, to obtain good performance of UIPS. This is because:

- $\eta_2$ acts like a capping threshold to ensure $\phi_{\boldsymbol{x},a}^* \le 2\eta_2$ holds even with small propensity scores. Hence, it should be set to a large value (e.g., 100).

- The key component (i.e., the first term) of $\phi_{\boldsymbol{x},a}^*$ can be rewritten in the following way. While all $(\boldsymbol{x}, a)$ pairs will be multiplied by $\phi_{\boldsymbol{x},a}^*$, $\eta_1$ in the numerator will not affect final performance too much, and the key is to find a good value of $\eta_1^2/\lambda$ to balance the two terms in the denominator:

$$\eta_1 / \left[ \exp(-\gamma U_{\boldsymbol{x},a}) + \frac{\eta_1^2/\lambda \cdot \pi_{\boldsymbol{\vartheta}}(a|\boldsymbol{x})^2}{\hat{\beta}(a|\boldsymbol{x})^2 \exp(-\gamma U_{\boldsymbol{x},a})} \right].$$

Thus with $\eta_1$ and $\eta_2$ fixed, effect of hyperparameter $\gamma$ and $\lambda$ on precision and recall can be found in Figure 2a and Figure 2b respectively.

### A.2.2 REAL-WORLD DATA

**Statistics of data.** The statistics of three real-world recommendation datasets with unbiased data can be found in Table 7.

All these datasets contain a set of biased data collected from users' interactions on the platform, and a set of unbiased data collected from a randomized controlled trial where items are randomly selected. As in (Ding et al. (2022)), on each dataset, the biased data is used for training, and the

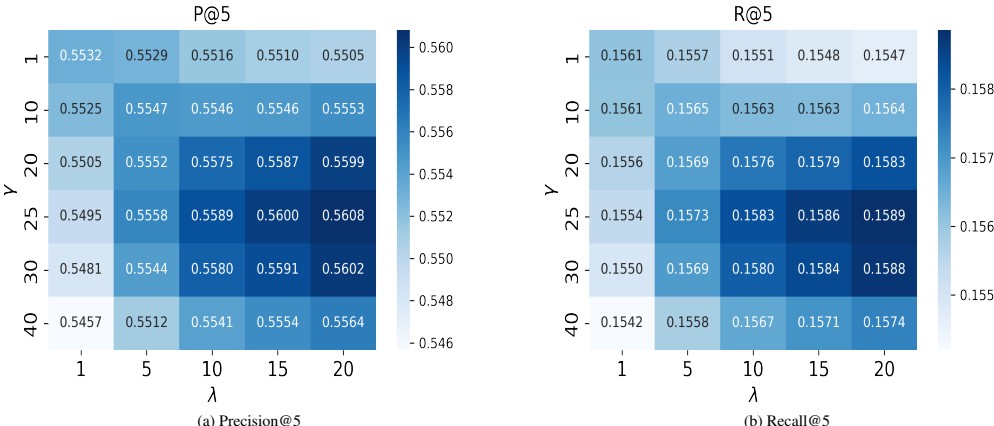

Figure 2: Effect of $\lambda$ and $\gamma$ on Precision@5 and Recall@5.

| Dataset | #User | #Item | #Biased Data | #Unbiased Data |
|---------|-------|-------|--------------|----------------|
| Yahoo | 15,400 | 1,000 | 311,704 | 54,000 |
| Coat | 290 | 300 | 6,960 | 4,640 |
| KuaiRec | 7,176 | 10,729 | 12,530,806 | 4,676,570 |

Table 7: The statistics of three real-world datasets.

unbiased data is for testing, with a small part of unbiased data split for validation purpose (5% on Yahoo and Coat, and 15% on KuaiRec). We take the reward as 1 if : (1) the rating is larger than 3 in Yahoo!R3 and Coat datasets; (2) the user watched more than 70% of the video in KuaiRec. Otherwise, the reward is labeled as 0.

**Implementation Details.** We adopt a two-tower neural network architecture to implement both the logging and learning policy, as shown in Figure 3. For the learning policy, the user representation and the item representation are first modelled through two separate neural networks (i.e., the user tower and the item tower), and then their element-by-element product vector is projected to predict the user's preference for the item. We then re-use the user state generated from the user tower of the learning policy, and model the logging policy with another separate item tower, following (Chen et al. (2019)). We also block gradients to prevent the logging policy interfering the user state of the learning policy. In each learning epoch, we will first estimate the logging policy, and then take the estimated logging probabilities as well as their uncertainties to optimize the learning policy. All hyperparameters are searched in a similar way described in Section 4.1.

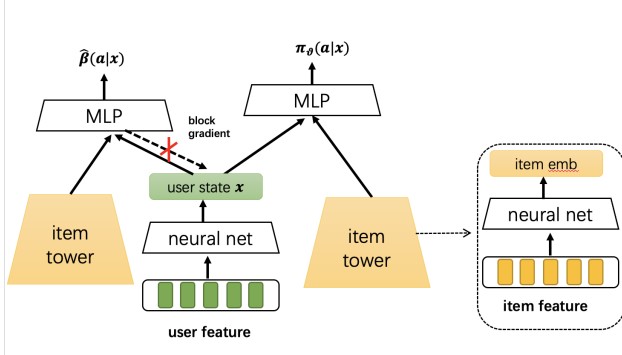

Figure 3: Model architecture of the logging and the learning policy in real-world datasets

## A.3 EXPERIMENTS ON THE DOUBLY ROBUST ESTIMATORS.

The doubly robust (DR) estimator (Jiang & Li (2016)), which is a hybrid of *direct method* (DM) estimator and *inverse propensity score* (IPS) estimator, is also widely used for off-policy evaluation. More specifically, let $\hat{\eta} : \mathcal{X} \times \mathcal{A} \to R$ be the imputation model in DM that estimates the reward of action $a$ under context vector $\boldsymbol{x}$, and $\hat{\beta}(a|\boldsymbol{x})$ be the estimated logging policy in the IPS estimator. The DR estimator evaluates the policy $\pi$ based on the logged dataset $D := \{(\boldsymbol{x}_n, a_n, r_{\boldsymbol{x}_n, a_n}) | n \in [N]\}$, by:

$$\hat{V}_{\mathrm{DR}}(\pi) = \hat{V}_{\mathrm{DM}}(\pi) + \frac{1}{N} \sum_{n=1}^{N} \frac{\pi(a_n|\boldsymbol{x}_n)}{\hat{\beta}(a_n|\boldsymbol{x}_n)} \left( r_{\boldsymbol{x}_n, a_n} - \hat{\eta}(\boldsymbol{x}_n, a_n) \right) \tag{11}$$

where $\hat{V}_{\mathrm{DM}}(\pi)$ is the DM estimator:

$$\hat{V}_{\mathrm{DM}}(\pi) = \frac{1}{N} \sum_{n=1}^{N} \sum_{a \in \mathcal{A}} \pi(a|\boldsymbol{x}_n)\hat{\eta}(\boldsymbol{x}_n, a). \tag{12}$$

Again assuming the policy $\pi(a|\boldsymbol{x})$ is parameterized by $\boldsymbol{\vartheta}$, the REINFORCE gradient of $\hat{V}_{\mathrm{DR}}(\pi_{\boldsymbol{\vartheta}})$ with respect to $\boldsymbol{\vartheta}$ can be readily derived as follows:

$$\nabla_{\boldsymbol{\vartheta}} \hat{V}_{\mathrm{DR}}(\pi_{\boldsymbol{\vartheta}}) = \frac{1}{N} \sum_{n=1}^{N} \left( \sum_{a \in \mathcal{A}} \pi_{\boldsymbol{\vartheta}}(a|\boldsymbol{x}_n)\hat{\eta}(\boldsymbol{x}_n, a)\nabla_{\boldsymbol{\vartheta}} \log(\pi_{\boldsymbol{\vartheta}}(a|\boldsymbol{x}_n)) \right)$$
$$+ \frac{1}{N} \sum_{n=1}^{N} \left( \frac{\pi(a_n|\boldsymbol{x}_n)}{\hat{\beta}(a_n|\boldsymbol{x}_n)}(r_{\boldsymbol{x}_n, a_n} - \hat{\eta}(\boldsymbol{x}_n, a_n)\nabla_{\boldsymbol{\vartheta}} \log(\pi_{\boldsymbol{\vartheta}}(a_n|\boldsymbol{x}_n)) \right). \tag{13}$$

The imputation model $\hat{\eta}(\boldsymbol{x}, a)$ is pre-trained following previous work (Liu et al. (2022)) with the same model architecture as the logging policy model. Besides the standard DR estimator, we also adapt UIPS and the best two baselines on off-policy evalaution estimator (i.e., MinVar and Shrinkage) to doubly robust setting using the same imputation model.

Table 8 and Table 9 show the results on the synthetic datasets and three real-world datasets respectively. For ease of comparison, we also include the experimental results of IPS-Cap and UIPS on each dataset in two tables. Two p-values are also provided: (1) P-value(UIPSDR): The p-value under the t-test between UIPSDR and the best DR baseline on each dataset; (2) P-value(UIPS): The p-value under the t-test between UIPS and the best DR baseline on each dataset. From Table 8 and Table 9, we can first observe that DR cannot consistently outperform IPS-Cap: It outperforms IPS-Cap on the Coat and KuaiRec dataset, while achieving much worse performance on the synthetic datasets and Yahoo dataset. This is because the imputation model also plays an important role in gradient calculation as shown in Equation(13), so its accuracy greatly affects policy learning. When the imputation model is sufficiently accurate, for example, on the Coat dataset with only 300 actions, incorporating the DM estimator not only leads to better performance of DR over IPS, but also improved performance of UIPSDR over UIPS. And in particular, in this situation UIPSDR performs better than DR with the gain being statistically significant. When the imputation model is not accurate enough, for example, on the KuaiRec dataset with a large action space but sparse reward feedback, DR is still worse than UIPS, and UIPSDR also performs worse than UIPS due to the distortion of the imputation model.

## A.4 THEORETICAL PROOF.

**Proof of Proposition 1:**

*Proof.* With the linearity of expectation, we have $\mathbb{E}_D \left[ \hat{V}_{\mathrm{BIPS}}(\pi_{\boldsymbol{\vartheta}}) \right] = \mathbb{E}_{\beta^*} \left[ \frac{\pi_{\boldsymbol{\vartheta}}(a|\boldsymbol{x})}{\hat{\beta}(a|\boldsymbol{x})} r_{\boldsymbol{x},a} \right]$, thus:

$$\mathrm{Bias} \left( \hat{\mathrm{V}}_{\mathrm{BIPS}}(\pi_{\boldsymbol{\vartheta}}) \right) = \mathbb{E}_D \left[ \hat{V}_{\mathrm{BIPS}}(\pi_{\boldsymbol{\vartheta}}) - V(\pi_{\boldsymbol{\vartheta}}) \right]$$
$$= \mathbb{E}_{\beta^*} \left[ \frac{\pi_{\boldsymbol{\vartheta}}(a|\boldsymbol{x})}{\hat{\beta}(a|\boldsymbol{x})} r_{\boldsymbol{x},a} \right] - \mathbb{E}_{\beta^*} \left[ \frac{\pi_{\boldsymbol{\vartheta}}(a|\boldsymbol{x})}{\beta^*(a|\boldsymbol{x})} r_{\boldsymbol{x},a} \right]$$
$$= \mathbb{E}_{\beta^*} \left[ \frac{\pi_{\boldsymbol{\vartheta}}(a|\boldsymbol{x})}{\beta^*(a|\boldsymbol{x})} r_{\boldsymbol{x},a} \left( \frac{\beta^*(a|\boldsymbol{x})}{\hat{\beta}(a|\boldsymbol{x})} - 1 \right) \right]$$
$$= \mathbb{E}_{\pi_{\boldsymbol{\vartheta}}} \left[ r_{\boldsymbol{x},a} \left( \frac{\beta^*(a|\boldsymbol{x})}{\hat{\beta}(a|\boldsymbol{x})} - 1 \right) \right]. \tag{14}$$

| | $\tau = 0.5$ | | | $\tau = 1$ | | | $\tau = 2$ | | |
|---|---|---|---|---|---|---|---|---|---|
| Algorithm | P@5 | R@5 | NDCG@5 | P@5 | R@5 | NDCG@5 | P@5 | R@5 | NDCG@5 |
| IPS-Cap | $0.5515{\pm}2e^{-3}$ | $0.1553{\pm}8e^{-4}$ | $0.6031{\pm}2e^{-3}$ | $0.5526{\pm}2e^{-3}$ | $0.1561{\pm}6e^{-4}$ | $0.6016{\pm}1e^{-3}$ | $0.5409{\pm}3e^{-3}$ | $0.1529{\pm}9e^{-4}$ | $0.5901{\pm}2e^{-3}$ |
| UIPS | $0.5589{\pm}3e^{-3}$ | $0.1583{\pm}9e^{-4}$ | $0.6095{\pm}3e^{-3}$ | $0.5572{\pm}2e^{-3}$ | $0.1571{\pm}8e^{-4}$ | $0.6074{\pm}2e^{-3}$ | $0.5432{\pm}3e^{-3}$ | $0.1534{\pm}8e^{-4}$ | $0.5946{\pm}2e^{-3}$ |
| DR | $0.3846{\pm}3e^{-2}$ | $0.1082{\pm}8e^{-3}$ | $0.4684{\pm}3e^{-2}$ | $0.3631{\pm}3e^{-2}$ | $0.1017{\pm}9e^{-3}$ | $0.4494{\pm}3e^{-2}$ | $0.3560{\pm}3e^{-2}$ | $0.0995{\pm}7e^{-3}$ | $0.4470{\pm}2e^{-2}$ |
| MinVarDR | $0.3212{\pm}3e^{-2}$ | $0.0908{\pm}8e^{-3}$ | $0.4062{\pm}3e^{-2}$ | $0.3240{\pm}5e^{-2}$ | $0.0903{\pm}1e^{-2}$ | $0.3905{\pm}5e^{-2}$ | $0.3234{\pm}5e^{-2}$ | $0.0910{\pm}1e^{-2}$ | $0.4059{\pm}4e^{-2}$ |
| ShrinkageDR | $0.4139{\pm}2e^{-2}$ | $0.1161{\pm}7e^{-3}$ | $0.4969{\pm}3e^{-2}$ | $0.3944{\pm}3e^{-2}$ | $0.1101{\pm}8e^{-3}$ | $0.4797{\pm}2e^{-2}$ | $0.4080{\pm}3e^{-2}$ | $0.1135{\pm}7e^{-3}$ | $0.4901{\pm}2e^{-2}$ |
| UIPSDR | $\mathbf{0.4278{\pm}2e^{-2}}$ | $\mathbf{0.1200{\pm}6e^{-3}}$ | $\mathbf{0.5069{\pm}2e^{-2}}$ | $\mathbf{0.4008{\pm}2e^{-2}}$ | $\mathbf{0.1126{\pm}7e^{-3}}$ | $\mathbf{0.4847{\pm}2e^{-2}}$ | $\mathbf{0.4144{\pm}2e^{-2}}$ | $\mathbf{0.1162{\pm}8e^{-3}}$ | $\mathbf{0.4972{\pm}2e^{-2}}$ |
| P-value(UIPSDR) | $2e^{-1}$ | $2e^{-1}$ | $3e^{-1}$ | $6e^{-1}$ | $4e^{-1}$ | $6e^{-1}$ | $6e^{-1}$ | $4e^{-1}$ | $5e^{-1}$ |
| P-value(UIPS) | $6e^{-13}$ | $4e^{-13}$ | $4e^{-12}$ | $2e^{-12}$ | $1e^{-12}$ | $5e^{-12}$ | $8e^{-12}$ | $8e^{-12}$ | $2e^{-11}$ |

Table 8: Experiment results on synthetic datasets. The best and second best results are highlighted with **bold** and underline respectively. Two p-values are calculated: (1) P-value(UIPSDR): The p-value under the t-test between UIPSDR and the best DR baseline on each dataset; (2) P-value(UIPS): The p-value under the t-test between UIPS and the best DR baseline on each dataset.

| | Yahoo | | | Coat | | | KuaiRec | | |
|---|---|---|---|---|---|---|---|---|---|
| Algorithm | P@5 | R@5 | NDCG@5 | P@5 | R@5 | NDCG@5 | P@50 | R@50 | NDCG@50 |
| IPS-Cap | $0.2751{\pm}2e^{-3}$ | $0.7419{\pm}8e^{-3}$ | $0.5928{\pm}7e^{-3}$ | $0.2758{\pm}6e^{-3}$ | $0.4582{\pm}7e^{-3}$ | $0.4399{\pm}9e^{-3}$ | $0.8750{\pm}3e^{-3}$ | $0.0238{\pm}7e^{-5}$ | $0.8788{\pm}5e^{-3}$ |
| UIPS | $0.2868{\pm}2e^{-3}$ | $0.7742{\pm}5e^{-3}$ | $0.6274{\pm}5e^{-3}$ | $0.2877{\pm}3e^{-3}$ | $0.4757{\pm}5e^{-3}$ | $0.4576{\pm}8e^{-3}$ | $0.9120{\pm}1e^{-3}$ | $0.0250{\pm}5e^{-5}$ | $0.9174{\pm}7e^{-4}$ |
| DR | $0.2670{\pm}2e^{-3}$ | $0.7174{\pm}6e^{-3}$ | $0.5636{\pm}6e^{-3}$ | $0.2884{\pm}3e^{-3}$ | $0.4760{\pm}5e^{-3}$ | $0.4541{\pm}5e^{-3}$ | $0.8794{\pm}1e^{-2}$ | $0.0240{\pm}5e^{-4}$ | $0.8824{\pm}2e^{-2}$ |
| MinVarDR | $0.2272{\pm}5e^{-3}$ | $0.5989{\pm}1e^{-2}$ | $0.4525{\pm}1e^{-2}$ | $0.2704{\pm}4e^{-3}$ | $0.4434{\pm}9e^{-3}$ | $0.4137{\pm}6e^{-3}$ | $0.8640{\pm}7e^{-3}$ | $0.0235{\pm}2e^{-4}$ | $0.8657{\pm}7e^{-3}$ |
| ShrinkageDR | $0.2697{\pm}2e^{-3}$ | $0.7226{\pm}6e^{-3}$ | $0.5713{\pm}5e^{-3}$ | $0.2895{\pm}4e^{-3}$ | $0.4749{\pm}6e^{-3}$ | $0.4526{\pm}6e^{-3}$ | $0.8778{\pm}2e^{-2}$ | $0.0239{\pm}5e^{-4}$ | $0.8800{\pm}2e^{-2}$ |
| UIPSDR | $\mathbf{0.2721{\pm}1e^{-3}}$ | $\mathbf{0.7294{\pm}6e^{-3}}$ | $\mathbf{0.5750{\pm}5e^{-3}}$ | $\mathbf{0.2946{\pm}4e^{-3}}$ | $\mathbf{0.4854{\pm}8e^{-3}}$ | $\mathbf{0.4647{\pm}8e^{-3}}$ | $\mathbf{0.8849{\pm}1e^{-2}}$ | $\mathbf{0.0242{\pm}4e^{-4}}$ | $\mathbf{0.8896{\pm}1e^{-2}}$ |
| P-value(UIPSDR) | $1e^{-2}$ | $2e^{-2}$ | $1e^{-1}$ | $7e^{-3}$ | $5e^{-3}$ | $2e^{-3}$ | $4e^{-1}$ | $4e^{-1}$ | $3e^{-1}$ |
| P-value(UIPS) | $1e^{-12}$ | $6e^{-14}$ | $6e^{-15}$ | $3e^{-1}$ | $8e^{-1}$ | $1e^{-1}$ | $2e^{-6}$ | $2e^{-6}$ | $1e^{-3}$ |

Table 9: Experimental results on real-world unbiased datasets. The best and second best results are highlighted with **bold** and underline respectively. Two p-values are calculated: (1) P-value(UIPSDR): The p-value under the t-test between UIPSDR and the best DR baseline on each dataset; (2) P-value(UIPS): The p-value under the t-test between UIPS and the best DR baseline on each dataset.

For variance, since samples are independently sampled from logging policy, thus :

$$\mathrm{Var}_D\left(\hat{V}_{\mathrm{BIPS}}(\pi_{\boldsymbol{\vartheta}})\right) = \frac{1}{N}\mathrm{Var}_{\beta^*}\left(\frac{\pi_{\boldsymbol{\vartheta}}(a|\boldsymbol{x})}{\hat{\beta}(a|\boldsymbol{x})}r_{\boldsymbol{x},a}\right).$$

By re-scaling, we get:

$$
\begin{aligned}
N \cdot \mathrm{Var}_D\left(\hat{V}_{\mathrm{BIPS}}(\pi_{\boldsymbol{\vartheta}})\right) &= \mathrm{Var}_{\beta^*}\left(\frac{\pi_{\boldsymbol{\vartheta}}(a|\boldsymbol{x})}{\hat{\beta}(a|\boldsymbol{x})}r_{\boldsymbol{x},a}\right) \\
&= \mathbb{E}_{\beta^*}\left[\frac{\pi_{\boldsymbol{\vartheta}}(a|\boldsymbol{x})^2}{\hat{\beta}(a|\boldsymbol{x})^2}r_{\boldsymbol{x},a}^2\right] - \left(\mathbb{E}_{\beta^*}\left[\frac{\pi_{\boldsymbol{\vartheta}}(a|\boldsymbol{x})}{\hat{\beta}(a|\boldsymbol{x})}r_{\boldsymbol{x},a}\right]\right)^2 \\
&= \mathbb{E}_{\pi_{\boldsymbol{\vartheta}}}\left[\frac{\pi_{\boldsymbol{\vartheta}}(a|\boldsymbol{x})}{\beta^*(a|\boldsymbol{x})} \cdot \frac{\beta^*(a|\boldsymbol{x})^2}{\hat{\beta}(a|\boldsymbol{x})^2}r_{\boldsymbol{x},a}^2\right] - \left(\mathbb{E}_{\pi_{\boldsymbol{\vartheta}}}\left[\frac{\beta^*(a|\boldsymbol{x})}{\hat{\beta}(a|\boldsymbol{x})}r_{\boldsymbol{x},a}\right]\right)^2 \\
&= \mathrm{Var}_{\pi_{\boldsymbol{\vartheta}}}\left(\frac{\beta^*(a|\boldsymbol{x})}{\hat{\beta}(a|\boldsymbol{x})}r_{\boldsymbol{x},a}\right) + \mathbb{E}_{\pi_{\boldsymbol{\vartheta}}}\left[\left(\frac{\pi_{\boldsymbol{\vartheta}}(a|\boldsymbol{x})}{\beta^*(a|\boldsymbol{x})} - 1\right) \cdot \frac{\beta^*(a|\boldsymbol{x})^2}{\hat{\beta}(a|\boldsymbol{x})^2}r_{\boldsymbol{x},a}^2\right]. \quad (15)
\end{aligned}
$$

Then we complete the proof. $\qquad\square$

**Proof of Theorem 1:**

*Proof.* We can get:

$$
\begin{aligned}
\mathrm{MSE}\left(\hat{V}_{\mathrm{UIPS}}(\pi_{\boldsymbol{\vartheta}})\right) &= \mathbb{E}_D\left[\left(\hat{V}_{\mathrm{UIPS}}(\pi_{\boldsymbol{\vartheta}}) - V(\pi_{\boldsymbol{\vartheta}})\right)^2\right] \\
&= \left(\mathbb{E}_D\left[\hat{V}_{\mathrm{UIPS}}(\pi_{\boldsymbol{\vartheta}}) - V(\pi_{\boldsymbol{\vartheta}})\right]\right)^2 + \mathrm{Var}_D\left(\hat{V}_{\mathrm{UIPS}}(\pi_{\boldsymbol{\vartheta}}) - V(\pi_{\boldsymbol{\vartheta}})\right) \\
&= \left(\mathbb{E}_D\left[\hat{V}_{\mathrm{UIPS}}(\pi_{\boldsymbol{\vartheta}}) - V(\pi_{\boldsymbol{\vartheta}})\right]\right)^2 + \mathrm{Var}_D\left(\hat{V}_{\mathrm{UIPS}}(\pi_{\boldsymbol{\vartheta}})\right) \\
&= \mathrm{Bias}(\hat{V}_{\mathrm{UIPS}}(\pi_{\boldsymbol{\vartheta}}))^2 + \mathrm{Var}(\hat{V}_{\mathrm{UIPS}}(\pi_{\boldsymbol{\vartheta}})).
\end{aligned}
$$

We first bound the bias term:

$$\text{Bias}(\hat{V}_{\text{UIPS}}(\pi_{\vartheta})) = \mathbb{E}_D\left[\hat{V}_{\text{UIPS}}(\pi_{\vartheta}) - V(\pi_{\vartheta})\right]$$

$$= \mathbb{E}_{\beta^*}\left[\frac{\pi_{\vartheta}(a|\boldsymbol{x})}{\hat{\beta}(a|\boldsymbol{x})}\phi_{\boldsymbol{x},a}r_{\boldsymbol{x},a}\right] - V(\pi_{\vartheta}) \tag{1}$$

$$= \mathbb{E}_{\beta^*}\left[\frac{\pi_{\vartheta}(a|\boldsymbol{x})}{\hat{\beta}(a|\boldsymbol{x})}\phi_{\boldsymbol{x},a}r_{\boldsymbol{x},a} - \frac{\pi_{\vartheta}(a|\boldsymbol{x})}{\beta^*(a|\boldsymbol{x})}r_{\boldsymbol{x},a}\right]$$

$$= \mathbb{E}_{\beta^*}\left[r_{\boldsymbol{x},a}\frac{\pi_{\vartheta}(a|\boldsymbol{x})}{\beta^*(a|\boldsymbol{x})}\cdot\left(\frac{\beta^*(a|\boldsymbol{x})}{\hat{\beta}(a|\boldsymbol{x})}\phi_{\boldsymbol{x},a} - 1\right)\right]$$

$$\leq \sqrt{\mathbb{E}_{\pi_{\vartheta}}\left[r_{\boldsymbol{x},a}^2\frac{\pi_{\vartheta}(a|\boldsymbol{x})}{\beta^*(a|\boldsymbol{x})}\right]}\cdot\sqrt{\mathbb{E}_{\beta^*}\left[\left(\frac{\beta^*(a|\boldsymbol{x})}{\hat{\beta}(a|\boldsymbol{x})}\phi_{\boldsymbol{x},a} - 1\right)^2\right]} \tag{2}$$

Equality (1) follows the linearity of expectation. Inequality (2) is due to the Cauchy-Schwarz inequality. We then bound the variance term:

$$\text{Var}(\hat{V}_{\text{UIPS}}(\pi_{\vartheta})) = \frac{1}{N}\text{Var}_{\beta^*}\left(\frac{\pi_{\vartheta}(a|\boldsymbol{x})}{\hat{\beta}(a|\boldsymbol{x})}\phi_{\boldsymbol{x},a}r_{\boldsymbol{x},a}\right) \tag{1}$$

$$= \frac{1}{N}\left(\mathbb{E}_{\beta^*}\left[\frac{\pi_{\vartheta}(a|\boldsymbol{x})^2}{\hat{\beta}(a|\boldsymbol{x})^2}\phi_{\boldsymbol{x},a}^2r_{\boldsymbol{x},a}^2\right] - \left(\mathbb{E}_{\beta^*}\left[\frac{\pi_{\vartheta}(a|\boldsymbol{x})}{\hat{\beta}(a|\boldsymbol{x})}\phi_{\boldsymbol{x},a}r_{\boldsymbol{x},a}\right]\right)^2\right) \tag{16}$$

$$\leq \frac{1}{N}\mathbb{E}_{\beta^*}\left[\frac{\pi_{\vartheta}(a|\boldsymbol{x})^2}{\hat{\beta}(a|\boldsymbol{x})^2}\phi_{\boldsymbol{x},a}^2r_{\boldsymbol{x},a}^2\right] \leq \mathbb{E}_{\beta^*}\left[\frac{\pi_{\vartheta}(a|\boldsymbol{x})^2}{\hat{\beta}(a|\boldsymbol{x})^2}\phi_{\boldsymbol{x},a}^2\right] \tag{17}$$

Combining the bound of bias and variance, we can complete the proof. $\square$

**Proof of Theorem 2:**

*Proof.* We first define several notations:

- $T(\phi_{\boldsymbol{x},a}, \beta_{\boldsymbol{x},a}) = \lambda\mathbb{E}_{\beta^*}\left[\left(\frac{\beta_{\boldsymbol{x},a}}{\hat{\beta}(a|\boldsymbol{x})}\phi_{\boldsymbol{x},a} - 1\right)^2\right] + \mathbb{E}_{\beta^*}\left[\frac{\pi_{\vartheta}(a|\boldsymbol{x})^2}{\hat{\beta}(a|\boldsymbol{x})^2}\phi_{\boldsymbol{x},a}^2\right]$.

- $\tilde{T}(\phi_{\boldsymbol{x},a}) = \max_{\beta_{\boldsymbol{x},a}\in\boldsymbol{B}_{\boldsymbol{x},a}}T(\phi_{\boldsymbol{x},a}, \beta_{\boldsymbol{x},a})$ denotes the maximum value of inner problem.

- $T^* = \min_{\phi_{\boldsymbol{x},a}}\tilde{T}(\phi_{\boldsymbol{x},a}) = \min_{\phi_{\boldsymbol{x},a}}\max_{\beta_{\boldsymbol{x},a}\in\boldsymbol{B}_{\boldsymbol{x},a}}T(\phi_{\boldsymbol{x},a}, \beta_{\boldsymbol{x},a})$ denote the optimal min-max value. And $\phi_{\boldsymbol{x},a}^* = \arg\min_{\phi_{\boldsymbol{x},a}}\tilde{T}(\phi_{\boldsymbol{x},a})$.

- $\boldsymbol{B}_{\boldsymbol{x},a}^- := \frac{\hat{Z}\exp(-\gamma U_{\boldsymbol{x},a})}{Z^*}\hat{\beta}(a|\boldsymbol{x})$, and $\boldsymbol{B}_{\boldsymbol{x},a}^+ := \frac{\hat{Z}\exp(\gamma U_{\boldsymbol{x},a})}{Z^*}\hat{\beta}(a|\boldsymbol{x})$.

We first find the maximum value of inner problem, i.e., $\tilde{T}(\phi_{\boldsymbol{x},a})$ for any fixed $\phi_{\boldsymbol{x},a}$. And there are three cases shown in Figure 4:

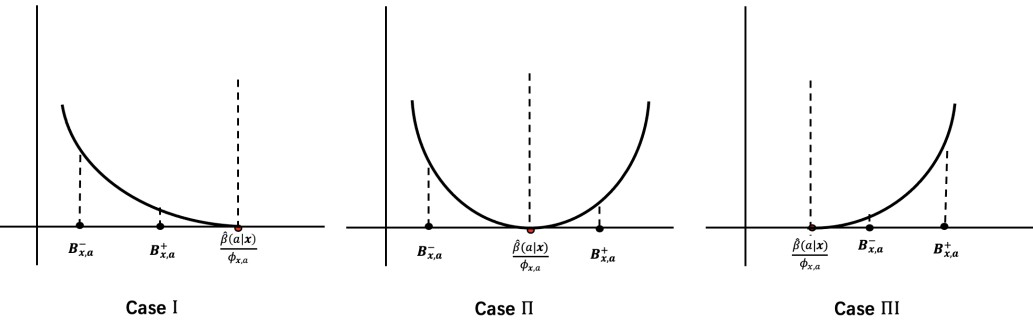

Figure 4: Three cases for maximizing inner problem.

**Case I:** When $\frac{\hat{\beta}(a|\boldsymbol{x})}{\phi_{\boldsymbol{x},a}} \geq \boldsymbol{B}_{\boldsymbol{x},a}^+$, , $\tilde{T}(\phi_{\boldsymbol{x},a})$ achieves the maximum value at $\beta_{\boldsymbol{x},a} = \boldsymbol{B}_{\boldsymbol{x},a}^-$. In other words, $\tilde{T}(\phi_{\boldsymbol{x},a}) = T(\phi_{\boldsymbol{x},a}, \boldsymbol{B}_{\boldsymbol{x},a}^-)$ when $\phi_{\boldsymbol{x},a} \leq \frac{\hat{\beta}(a|\boldsymbol{x})}{\boldsymbol{B}_{\boldsymbol{x},a}^+}$.

**Case II:** When $\boldsymbol{B}_{\boldsymbol{x},a}^{-} \leq \frac{\hat{\beta}(a|\boldsymbol{x})}{\phi_{\boldsymbol{x},a}} \leq \boldsymbol{B}_{\boldsymbol{x},a}^{+}$, i.e., $\frac{Z^* \exp(-\gamma U_{\boldsymbol{x},a})}{\hat{Z}} \leq \phi_{\boldsymbol{x},a} \leq \frac{Z^* \exp(\gamma U_{\boldsymbol{x},a})}{\hat{Z}}$, then $\tilde{T}(\phi_{\boldsymbol{x},a})$ will be the maximum between $T(\phi_{\boldsymbol{x},a}, \boldsymbol{B}_{\boldsymbol{x},a}^{-})$ and $T(\phi_{\boldsymbol{x},a}, \boldsymbol{B}_{\boldsymbol{x},a}^{+})$.

More specifically, when $\frac{\hat{\beta}(a|\boldsymbol{x})}{\phi_{\boldsymbol{x},a}} \leq \frac{\boldsymbol{B}_{\boldsymbol{x},a}^{+}+\boldsymbol{B}_{\boldsymbol{x},a}^{-}}{2}$, i.e., $\phi_{\boldsymbol{x},a} \geq \frac{2\hat{\beta}(a|\boldsymbol{x})}{\boldsymbol{B}_{\boldsymbol{x},a}^{+}+\boldsymbol{B}_{\boldsymbol{x},a}^{-}}$, $\tilde{T}(\phi_{\boldsymbol{x},a}) = T(\phi_{\boldsymbol{x},a}, \boldsymbol{B}_{\boldsymbol{x},a}^{+})$. Otherwise when $\phi_{\boldsymbol{x},a} < \frac{2\hat{\beta}(a|\boldsymbol{x})}{\boldsymbol{B}_{\boldsymbol{x},a}^{+}+\boldsymbol{B}_{\boldsymbol{x},a}^{-}}$, $\tilde{T}(\phi_{\boldsymbol{x},a}) = T(\phi_{\boldsymbol{x},a}, \boldsymbol{B}_{\boldsymbol{x},a}^{-})$.

**Case III:** When $\phi_{\boldsymbol{x},a} \geq \frac{\hat{\beta}(a|\boldsymbol{x})}{\boldsymbol{B}_{\boldsymbol{x},a}^{-}}$. implying $\frac{\hat{\beta}(a|\boldsymbol{x})}{\phi_{\boldsymbol{x},a}} \leq \boldsymbol{B}_{\boldsymbol{x},a}^{-}$, $\tilde{T}(\phi_{\boldsymbol{x},a}) = T(\phi_{\boldsymbol{x},a}, \boldsymbol{B}_{\boldsymbol{x},a}^{+})$.

Overall, we get that:

$$\tilde{T}(\phi_{\boldsymbol{x},a}) = \begin{cases} T(\phi_{\boldsymbol{x},a}, \boldsymbol{B}_{\boldsymbol{x},a}^{-}), & \phi_{\boldsymbol{x},a} \in (-\infty, \frac{2\hat{\beta}(a|\boldsymbol{x})}{\boldsymbol{B}_{\boldsymbol{x},a}^{+}+\boldsymbol{B}_{\boldsymbol{x},a}^{-}}] \\ T(\phi_{\boldsymbol{x},a}, \boldsymbol{B}_{\boldsymbol{x},a}^{+}) & \phi_{\boldsymbol{x},a} \in [\frac{2\hat{\beta}(a|\boldsymbol{x})}{\boldsymbol{B}_{\boldsymbol{x},a}^{+}+\boldsymbol{B}_{\boldsymbol{x},a}^{-}}, \infty) \end{cases} \tag{18}$$

Next we try to find the minimum value of $\tilde{T}(\phi_{\boldsymbol{x},a})$. We first observe that without considering constraint on $\phi_{\boldsymbol{x},a}$, when

$$\phi_{\boldsymbol{x},a}^{+} = \frac{\lambda}{\lambda \frac{\hat{Z}\exp(\gamma U_{\boldsymbol{x},a})}{Z^*} + \frac{\pi_{\vartheta}(a|\boldsymbol{x})^2}{\hat{\beta}(a|\boldsymbol{x})^2 \frac{\hat{Z}\exp(\gamma U_{\boldsymbol{x},a})}{Z^*}}},$$

$T(\phi_{\boldsymbol{x},a}, \boldsymbol{B}_{\boldsymbol{x},a}^{+})$ achieves the global minimum value. However, $\phi_{\boldsymbol{x},a}^{+} \leq \frac{\hat{\beta}(a|\boldsymbol{x})}{\boldsymbol{B}_{\boldsymbol{x},a}^{+}}$, which implies when $\phi_{\boldsymbol{x},a} \in [\frac{2\hat{\beta}(a|\boldsymbol{x})}{\boldsymbol{B}_{\boldsymbol{x},a}^{+}+\boldsymbol{B}_{\boldsymbol{x},a}^{-}}, \infty)$, the minimum value of $T(\phi_{\boldsymbol{x},a}, \boldsymbol{B}_{\boldsymbol{x},a}^{+})$ achieves at $\frac{2\hat{\beta}(a|\boldsymbol{x})}{\boldsymbol{B}_{\boldsymbol{x},a}^{+}+\boldsymbol{B}_{\boldsymbol{x},a}^{-}}$.

On the other hand, without considering any constraint on $\phi_{\boldsymbol{x},a}$, global minimum value of $T(\phi_{\boldsymbol{x},a}, \boldsymbol{B}_{\boldsymbol{x},a}^{-})$ achieves at:

$$\phi_{\boldsymbol{x},a}^{-} = \frac{\lambda}{\lambda \frac{\hat{Z}\exp(-\gamma U_{\boldsymbol{x},a})}{Z^*} + \frac{\pi_{\vartheta}(a|\boldsymbol{x})^2}{\hat{\beta}(a|\boldsymbol{x})^2 \frac{\hat{Z}\exp(-\gamma U_{\boldsymbol{x},a})}{Z^*}}}. \tag{19}$$

Thus if $\phi_{\boldsymbol{x},a}^{-} \leq \frac{2\hat{\beta}(a|\boldsymbol{x})}{\boldsymbol{B}_{\boldsymbol{x},a}^{+}+\boldsymbol{B}_{\boldsymbol{x},a}^{-}}$, $\phi_{\boldsymbol{x},a}^{*} = \phi_{\boldsymbol{x},a}^{-}$. Otherwise $\phi_{\boldsymbol{x},a}^{*} = \frac{2\hat{\beta}(a|\boldsymbol{x})}{\boldsymbol{B}_{\boldsymbol{x},a}^{+}+\boldsymbol{B}_{\boldsymbol{x},a}^{-}}$.

Overall,

$$\phi_{\boldsymbol{x},a}^{*} = \min\left( \frac{\lambda}{\lambda \frac{\hat{Z}\exp(-\gamma U_{\boldsymbol{x},a})}{Z^*} + \frac{\pi_{\vartheta}(a|\boldsymbol{x})^2}{\hat{\beta}(a|\boldsymbol{x})^2 \frac{\hat{Z}\exp(-\gamma U_{\boldsymbol{x},a})}{Z^*}}}, \frac{2}{\frac{\hat{Z}\exp(\gamma U_{\boldsymbol{x},a})}{Z^*} + \frac{\hat{Z}\exp(-\gamma U_{\boldsymbol{x},a})}{Z^*}} \right) \tag{20}$$

We use $\eta_1$ and $\eta_2$ to represent $\frac{Z^*}{\hat{Z}}$ in the two terms respectively. We can get

$$\phi_{\boldsymbol{x},a}^{*} = \min\left( \frac{\lambda}{\frac{\lambda}{\eta_1}\exp(-\gamma U_{\boldsymbol{x},a}) + \frac{\eta_1 \pi_{\vartheta}(a|\boldsymbol{x})^2}{\hat{\beta}(a|\boldsymbol{x})^2 \exp(-\gamma U_{\boldsymbol{x},a})}}, \frac{2\eta_2}{\exp(\gamma U_{\boldsymbol{x},a}) + \exp(-\gamma U_{\boldsymbol{x},a})} \right) \tag{21}$$

where $\eta_1, \eta_2 \in [\exp(-\gamma U_s^{\max}), \exp(\gamma U_s^{\max})]$, since $\hat{Z}\exp(-\gamma U_s^{\max}) \leq Z^* = \sum_{a'} \exp(f_{\theta^*}(a'|\boldsymbol{x})) \leq \hat{Z}\exp(U_s^{\max})$. Usually we set $\eta_1 \leq \eta_2$. We introduce two parameters since the scale of $\eta_1$ is closely related to the scale of $\lambda$, while the scale of $\eta_2$ is independent.

Then we complete the proof . $\qquad\square$

**Lemma 1.** *Assume* $\eta_1 \leq \eta_2$, *then with fixed* $\pi_{\vartheta}(a|\boldsymbol{x})$ *and* $\hat{\beta}(a|\boldsymbol{x})$, *and* $\alpha_{\boldsymbol{x},a} = \sqrt{\frac{\lambda}{2\eta_1\eta_2} - \frac{\lambda(1-\eta_1)}{\eta_1^2}\exp(-2\gamma U_{\boldsymbol{x},a})}$, *we have the following observations:*

- *If* $\frac{\pi_{\vartheta}(a|\boldsymbol{x})}{\hat{\beta}(a|\boldsymbol{x})} \leq \alpha_{\boldsymbol{x},a}$, $\phi_{\boldsymbol{x},a}^{*} = 2\eta_2/[\exp(\gamma U_{\boldsymbol{x},a}) + \exp(-\gamma U_{\boldsymbol{x},a})]$. *Otherwise* $\phi_{\boldsymbol{x},a}^{*} = \lambda/\left[\frac{\lambda}{\eta_1}\exp(-\gamma U_{\boldsymbol{x},a}) + \frac{\eta_1 \pi_{\vartheta}(a|\boldsymbol{x})^2}{\hat{\beta}^2(a|\boldsymbol{x})\exp(-\gamma U_{\boldsymbol{x},a})}\right]$. *In other words,* $\phi_{\boldsymbol{x},a}^{*} \leq 2\eta_2$ *always holds.*

- If $\frac{\pi_{\boldsymbol{\vartheta}}(a|\boldsymbol{x})}{\hat{\beta}(a|\boldsymbol{x})} \geq \frac{\sqrt{\lambda}}{\eta_1}$ , then $\phi_{s,a}^*$ will always decrease as $U_{\boldsymbol{x},a}$ increases.
- If $\alpha_{\boldsymbol{x},a} \leq \frac{\pi_{\boldsymbol{\vartheta}}(a|\boldsymbol{x})}{\hat{\beta}(a|\boldsymbol{x})} < \frac{\sqrt{\lambda}}{\eta_1} \exp(-\gamma U_{\boldsymbol{x},a})$, larger $U_{\boldsymbol{x},a}$ brings larger $\phi_{\boldsymbol{x},a}^*$. Otherwise $\phi_{\boldsymbol{x},a}^*$ still decreases as $U_{\boldsymbol{x},a}$ increases.

*Proof.* For the first observation, deriving

$$\frac{\lambda}{\frac{\lambda}{\eta_1} \exp\left(-\gamma U_{\boldsymbol{x},a}\right) + \frac{\eta_1 \pi_{\boldsymbol{\vartheta}}(a|\boldsymbol{x})^2}{\hat{\beta}^2(a|\boldsymbol{x}) \exp(-\gamma U_{\boldsymbol{x},a})}} \leq \frac{2\eta_2}{\exp\left(\gamma U_{\boldsymbol{x},a}\right) + \exp\left(-\gamma U_{\boldsymbol{x},a}\right)}$$

we can get the result.

For the second and third observation, since $\eta_1 \leq \eta_2$, then $\alpha_{\boldsymbol{x},a} \leq \frac{\sqrt{\lambda}}{\sqrt{2}\eta_1} \leq \frac{\sqrt{\lambda}}{\eta_1}$. Let $\mathcal{L}(u) = \frac{\lambda}{\eta_1} \exp(-\gamma u) + \frac{\eta_1 \pi_{\boldsymbol{\vartheta}}(a|\boldsymbol{x})^2}{\hat{\beta}(a|\boldsymbol{x})^2 \exp(-\gamma u)}$, we can have:

$$\nabla_u \mathcal{L}(u) = -\gamma \frac{\lambda}{\eta_1} \exp(-\gamma u) + \gamma \frac{\eta_1 \pi_{\boldsymbol{\vartheta}}(a|\boldsymbol{x})^2}{\hat{\beta}(a|\boldsymbol{x})^2} \exp(\gamma u)$$

By letting $\nabla_u \mathcal{L}(u) \geq 0$, we can get $u \geq \frac{1}{\gamma} \log\left(\frac{\sqrt{\lambda}\hat{\beta}(a|\boldsymbol{x})}{\eta_1 \pi_{\boldsymbol{\vartheta}}(a|\boldsymbol{x})}\right)$. This implies when $U_{\boldsymbol{x},a} \geq \frac{1}{\gamma} \log\left(\frac{\sqrt{\lambda}\hat{\beta}(a|\boldsymbol{x})}{\eta_1 \pi_{\boldsymbol{\vartheta}}(a|\boldsymbol{x})}\right)$, $\phi_{\boldsymbol{x},a}^*$ will decrease as $U_{\boldsymbol{x},a}$ increases. Otherwise as $U_{\boldsymbol{x},a}$ increases, $\phi_{\boldsymbol{x},a}^*$ also increases.

Specially, when $\frac{\pi_{\boldsymbol{\vartheta}}(a|\boldsymbol{x})}{\hat{\beta}(a|\boldsymbol{x})} \geq \frac{\sqrt{\lambda}}{\eta_1}$, $\log\left(\frac{\sqrt{\lambda}\hat{\beta}(a|\boldsymbol{x})}{\eta_1 \pi_{\boldsymbol{\vartheta}}(a|\boldsymbol{x})}\right) \leq 0$. Given $U_{\boldsymbol{x},a} \geq 0$, this implies that $\phi_{\boldsymbol{x},a}^*$ will always decrease as $U_{\boldsymbol{x},a}$ increases in this case.

Otherwise, when $\alpha_{\boldsymbol{x},a} \leq \frac{\pi_{\boldsymbol{\vartheta}}(a|\boldsymbol{x})}{\hat{\beta}(a|\boldsymbol{x})} \leq \frac{\sqrt{\lambda}}{\eta_1} \exp(-\gamma U_{\boldsymbol{x},a})$, with

$$\frac{\pi_{\boldsymbol{\vartheta}}(a|\boldsymbol{x})}{\hat{\beta}(a|\boldsymbol{x}) \exp(-\gamma U_{\boldsymbol{x},a})} \leq \frac{\sqrt{\lambda}}{\eta_1} \implies U_{\boldsymbol{x},a} \leq \frac{1}{\gamma} \log\left(\frac{\sqrt{\lambda}\hat{\beta}(a|\boldsymbol{x})}{\eta_1 \pi_{\boldsymbol{\vartheta}}(a|\boldsymbol{x})}\right).$$

larger $U_{\boldsymbol{x},a}$ implies larger $\phi_{\boldsymbol{x},a}$. Otherwise $\phi_{\boldsymbol{x},a}^*$ still decreases as $U_{\boldsymbol{x},a}$ increases. This completes the proof. $\square$

