# OpenReview forum: "Uncertainty-aware off policy learning"
_ICLR.cc/2023/Conference — Submitted to ICLR 2023_

### Official Review · Reviewer_4K6H · 2022-10-23

**Confidence:** 4
**Correctness:** 4
**Technical Novelty And Significance:** 2
**Empirical Novelty And Significance:** 3
**Recommendation:** 3

**Clarity, Quality, Novelty And Reproducibility:**

Clarity: The paper is very clear. Theorems are clearly stated without ambiguity. Empirical results are described with details so that readers can easily understand.


Quality: The theorems are clear and the proof looks correct to me, though I did not check all the proof details. The experimental result is good and extensive.

Novelty: The technical novelty is unclear. Specifically, (1) the existence and bias when the data distribution is poorly estimated is well-known; (2) similar IPS correction term is also considered in other works; (3) no theoretical guarantee for the proposed estimator (minor issue).

Reproducibility: The reproducibility is good. Experimental details are provided.


**Strength And Weaknesses:**

***Strength***:

1. The paper is written very clearly. The problem is well-formulated. The theorems are stated clearly and there is no ambiguity. The experimental settings and results are presented with details.

2. The experimental results are extensive. Various datasets and cases are studied. The results corroborate the proposed estimator.
---
***Weakness***:

1. My major concern is that the technical novelty is unclear.

(i). The existence of bias for the IPS estimator when the data distribution estimation is not accurate is a well-known result in the area of offline policy evaluation.

(ii). IPS-based estimator is quite thoroughly studied. This paper studies the single-step setting, while IPS-type estimator is already extensively studied in the case of dynamic decision making/reinforcement learning. For example, in the DICE line of work ([1-3]), a series of distribution correction estimators are proposed (Also, these works are not discussed). Therefore, the correction term in this paper does not seem novel.

(iii). The most significant technical contribution seems to be the objective function equation 8. However, this follows directly from the robust optimization as mentioned by this paper.

Overall, it is not clear what is the technical novelty and contribution of this paper.

2. There is no theoretical guarantee for the sample complexity of the proposed estimator. (minor issue)
---
[1] O. Nachum, Y. Chow, B. Dai, and L. Li: DualDICE: Behavior-agnostic estimation of discounted stationary distribution corrections. In Advances in Neural Information Processing Systems 32 (NeurIPS), spotlight, 2019

[2] R. Zhang, B. Dai, L. Li, and D. Schuurmans: GenDICE: Generalized offline estimation of stationary values. In the 8th International Conference on Learning Representations (ICLR), 2020

[3] B. Dai, O. Nachum, Y. Chow, L. Li, Cs. Szepesvari, D. Schuurmans: CoinDICE: Off-policy confidence interval estimation. In Advances in Neural Information Processing Systems 33 (NeurIPS), spotlight, 2020


**Summary Of The Paper:**

This paper studies off-policy learning when there is distribution shift between the data distribution and target distribution. The paper points out that the vanilla inverse propensity score (IPS) estimator suffers from large variance and bias when the estimated data distribution is inaccurate. To this end, this paper proposes to take into consideration the uncertainty in the logging probability estimation, and constructs an uncertainty-aware IPS estimator. A mathematical framework about how to solve for the
UIPS estimator is described. Experimental results on both synthetic data and real-world data show that the proposed estimator performs better than a few benchmark algorithms on recommendation tasks.


**Summary Of The Review:**

My current recommendation is reject.

The main reason is that the technical novelty is unclear to me. I did not see what the technical contribution of this paper is.

However, this paper does have good and quite extensive experimental results.

---

> ### Author Response · Authors · 2022-11-17
> **Response to Reviewer 4K6H**
>
> We thank the reviewer for pointing out the place that unfortunately caused the misunderstanding of our contribution.
>
>
> [Q1] **"IPS-based estimator is quite thoroughly studied. This paper studies the single-step setting, while IPS-type estimator is already extensively studied in the case of dynamic decision making/reinforcement learning. For example, in the DICE line of work ([1-3]), a series of distribution correction estimators are proposed (Also, these works are not discussed). Therefore, the correction term in this paper does not seem novel."**
>
> [A1] The motivation of the proposed UIPS is **fundamentally different** from the DICE line of work.
>
> - The DICE line of work aims to address the large variance of off-policy correction weight in multi-step reinforcement learning caused by a series of products of propensity scores. They replace the product-based off-policy correction weight by the estimated discounted stationary distribution ratios, **which are assumed to be the ground-truth without considering the accuracy of the estimation**. Moreover, the DICE line of work is **not applicable when the next state is unknown beforehand in each logged sample, such as in the contextual bandit setting.**
>
> - Our work is motivated by the observations that directly taking the estimated logging policy to approximate the ground-truth logging policy results in a biased estimator which is sensitive to those inaccurate and small estimated logging probabilities.
> Thus, **our key contribution is to explicitly model the uncertainty of the estimated logging policy**, i.e., the difference between the ground-truth logging probabilities and the estimated logging probabilities, and propose an uncertainty-aware off-policy estimator (UIPS) for more accurate off-policy evaluation and learning.
>
> More detailed comparisons between these two lines of work are as follows.
>
> **Motivation of the DICE line of work.** Let $\tau=(s_0,a_0,s_1, \ldots, s_T, a_T)$ denote a trajectory sampled according to the logging policy $\beta(\cdot|\cdot)$, where $s_t$ and $a_t$ denote the state and the taken action at time t respectively.
> Normally, the given policy $\pi(a|s)$ is evaluated based solely on the set of logged trajectories $\{\tau\}$ by:
> $$V(\pi) = (1-\kappa) \mathbb{E}\left[ \left( \prod_{t=0}^{T-1} \frac{\pi(a_t|s_t)}{\beta(a_t|s_t)} \right) \left (\sum_{t=0}^T \kappa^t r_{s_t, a_t} \right) \right],$$
> where $\kappa$ is the discount factor for future rewards.
> The off-policy correction weight $\left( \prod_{t=0}^{T-1} \frac{\pi(a_t|s_t)}{\beta(a_t|s_t)} \right)$ involves a series of products and suffers from  huge variance, which is  known as "the curse of horizon"[1]. To avoid exponential dependence on trajectory length, the DICE line of work proposes to directly estimate the stationary distribution over $(s,a)$ pairs of a policy, i.e., $d^{\pi}(s,a)$ and $d^{\beta}(s,a)$, and evaluate $\pi(a|s)$ by:
> $$ V(\pi) = \mathbb{E}_{(s,a) \sim d^{\beta}} \left[ \frac{d^{\pi}(s,a)}{d^{\beta}(s,a) }  r\_{s,a}\right].$$
>
> **To estimate $d^{\beta}(s,a)$ and $d^{\pi}(s,a)$, one need to know the next state explicitly**.
> In other words, each sample should be like $(s_t,a_t, r_{s_t, a_t}, s_{t+1})$.
>
> Moreover, they directly take the estimated stationary distribution as the ground-truth one, and **do not consider the correctness of the estimation**. How to factor the accuracy of the estimated stationary distribution into DICE type models for further improvement is also an interesting future work.
>
> **Motivation of UIPS.**  When the ground-truth logging policy $\beta^*(a|\boldsymbol{x})$ is unknown, previous work directly takes the estimated logging policy $\hat{\beta}(a|\boldsymbol{x})$ for off-policy correction. In other words, they use the estimator
>  $\mathbb{E}\_{\beta^*}\left[ \frac{\pi(a|\boldsymbol{x})}{\color{blue}\hat{\beta}(a|\boldsymbol{x})} \cdot r_{\boldsymbol{x},a}\right]$  to approximate $\mathbb{E}\_{\beta^*}\left[ \frac{\pi(a|\boldsymbol{x})}{\color{red}\beta^*(a|\boldsymbol{x})} \cdot r_{\boldsymbol{x},a}\right]$ to evaluate a given policy $\pi(a|\boldsymbol{x})$. We theoretically demonstrated that such an approximation results in a biased estimator which is sensitive to those inaccurate and small estimated logging probabilities. We then consider the uncertainty of $\hat{\beta}(a|\boldsymbol{x})$, i.e., $|\hat{\beta}(a|\boldsymbol{x}) -\beta^*(a|\boldsymbol{x})|$, and propose UIPS for improved off-policy learning.
>
>
>
>
> [1]Li, Lihong, Rémi Munos, and Csaba Szepesvári. "Toward minimax off-policy value estimation." Artificial Intelligence and Statistics. PMLR, 2015

---

### Official Review · Reviewer_NQXb · 2022-10-24

**Confidence:** 4
**Correctness:** 3
**Technical Novelty And Significance:** 3
**Empirical Novelty And Significance:** 2
**Recommendation:** 5

**Clarity, Quality, Novelty And Reproducibility:**

Clarity:
The paper is well-written and most of the parts are clear.

Quality:
The paper is a solid paper. The method is well-motivated and clearly explained. The idea is neat with good empirical performance, though it would be great if the authors could provide more evidence about the significance of the gain. Theoretically, it would be great to analyze both the evaluation and learning performance, in terms of the quality of the uncertainty estimate. Empirically, the paper contains comprehensive results over both synthetic and real-world experiments, as well as valuable ablation studies (though the ablation over $\eta_1$ and $\eta_2$ is missing)? One weakness is the number of hyper-parameters used is high, and it is well-known for off-policy problems, these hyper-parameters are hard to select. The paper does not provide a concrete method on this part, which might reduce the impact of the paper.

Novelty:
The idea proposed by adding additional weight to quantify the uncertainty in IPS estimate is novel and neat.

Reproducibility:
The code is included.

**Strength And Weaknesses:**

Strength:

1). The unknown propensities are very common for real-world problems, especially when the underlying system is complex and the propensities are hard to calculate. This paper proposes efficient off-policy evaluation/learning method under this realistic setting.

2). The paper is well-written and easy to follow.

3). The method is well-motivated, solid, and the idea about reweighing based on uncertainty is neat.

3). It has comprehensive ablation studies and empirical results, with both synthetic and real-world datasets.

Weakness:

1). The proposed method seems have several important hyper-parameters, that control the balance of bias and variance term, as well as the confidence bound. The methodology part does not give a clear guideline about how to choosing them. From empirical studies, it seems to be selected from the unbiased validation set? If I understand it correctly, it might be hard to get this validation set in real-world scenarios. Could the authors comment more on how these are selected for different experiments?

2). The uncertainty estimate is crucial for the uncertainty weight in Theorem 2. The NTK theory provides a nice quantification about the uncertainty for deep neural networks. However, I am concerned about the dimensionality of the gradient vector $g$, as well as the inversion stability of the matrix M. What are their corresponding dimensionality for the experiments? [1] uses the inverse of the diagonal of the matrix M as approximations. Could the authors comment more on how the quality of the uncertainty estimate affects both the evaluation and learning? It should be reflected in the MSE of the estimator as well as the policy optimization regret bound.

3). All the results are average results, without standard deviation. It would be great if the std is provided, and this could provide us more information about whether the gain over previous method is significant.

4). Could the authors comment more on the convergence criteria? For the policy optimization step, is it performed until full convergence? This part seems missing from the paper, and it is worth commenting more about the computational cost as well.



Reference:
[1]. Neural Contextual Bandits with UCB-based Exploration


**Summary Of The Paper:**

This paper studies off-policy evaluation and learning, under unknown propensities, which is pretty common for most complicated systems. The paper proposes a novel Uncertainty-aware Inverse Propensity Score estimator, which builds upon the original IPS estimator by adding an additional weight to reflect the uncertainty of the estimated propensity score. This specific weight is derived by optimizing an upper bound of the MSE of the estimator. Empirical results on synthetic and real-world dataset shows better average performance compared to other baselines, in terms of MSE (for evaluation), Precision@K, Recall@K, etc (for learning).

**Summary Of The Review:**

I like the topic this paper studies, which seems to be relevant to lots of practitioners who work on OPE/OPL problems. This paper proposes a novel uncertainty-based IPS estimator, which is clearly motivated. Some minor theoretical guarantee of the proposed estimator and learning performance is missing. The empirical results are comprehensive, however it is unclear the significance of the gain, as well as the applicability of the method, due to hyper-parameter tuning as well as the training cost. Based on the current version of the paper, it is around the borderline from my perspective, but I would consider changing the score based on the authors' reply.

---

> ### Author Response · Authors · 2022-11-17
> **Response to Reviewer NQXb (part 2/2)**
>
> [Q2] **"I am concerned about the dimensionality of the gradient vector g, as well as the inversion stability of the matrix M. What are their corresponding dimensionality for the experiments?  Could the authors comment more on how the quality of the uncertainty estimate affects both the evaluation and learning? It should be reflected in the MSE of the estimator as well as the policy optimization regret bound.”"**
>
> [A2] Thanks for pointing out the place that could cause unnecessary confusion. We first clarify that the gradient vector $g$ is **the gradient of $f_{\theta}(\boldsymbol{x},a)$ with respect to its last layer, rather than to the entire network** [4]. We have revised the paper to make it clearer. In our experiments, the dimension of the gradient vector $g$ (i.e., the last linear layer) is 64 and 128 on synthetic and real-world datasets respectively.
>
> The proposed UIPS is built on the high probability confidence bound of a neural network developed in [4]. In practice, one can also use other estimators for the logging probability, as long as the confidence interval of the estimation can be easily (or analytically) obtained. We chose neural networks as an example to demonstrate UIPS, due to the strong representation learning power of neural networks. But how to derive more accurate uncertainty estimation of a neural network is beyond the scope of this work. And we also agree that inspecting the impact of confidence interval estimation on the quality of the policy evaluation as well as the resulting policy optimization in UIPS is an important future direction.
>
>
> [4] Xu, Pan, et al. "Neural Contextual Bandits with Deep Representation and Shallow Exploration." ICLR2021.
>
> [Q3] **"All the results are average results, without standard deviation. It would be great if the std is provided, and this could provide us more information about whether the gain over previous method is significant."**
>
> [A3] Please find our answer to CQ1 in the general response to all reviewers.
>
>
> [Q4] **"Could the authors comment more on the convergence criteria? For the policy optimization step, is it performed until full convergence? This part seems missing from the paper, and it is worth commenting more about the computational cost as well."**
>
> [A4] We used the performance on the validation dataset as the stopping criteria of our policy optimization. For the computation complexity, Please find our answer to CQ2 in the general response to all reviewers.

---

> ### Author Response · Authors · 2022-11-17
> **Response to Reviewer NQXb (part 1/2)**
>
> We thank the reviewer for constructive suggestions to clarify important arguments and enhance the experiment design and the proposed method's applicability, which significantly helped us strengthen our paper.
>
> [Q1] **"The proposed method seems have several important hyper-parameters, that control the balance of bias and variance term, as well as the confidence bound. The methodology part does not give a clear guideline about how to choosing them. From empirical studies, it seems to be selected from the unbiased validation set? If I understand it correctly, it might be hard to get this validation set in real-world scenarios. Could the authors comment more on how these are selected for different experiments?"**
>
> [A1] **Hyperparameter tuning.** We first want to remark that although UIPS has four hyperparameters ($\lambda$, $\gamma$, $\eta_1$, and $\eta_2$), one only needs to carefully finetune two of them, i.e., $\gamma$ and $\eta_1^2/\lambda$, to obtain good performance of UIPS. This is because:
> - $\eta_2$ acts like a capping threshold to ensure $\phi_{\boldsymbol{x},a}^* \leq 2 \eta_2$ holds even with small propensity scores. Hence, it should be set to a large value (e.g., 100).
> - The key component (i.e., the first term) of $ \phi_{\boldsymbol{x},a}^* $ can be rewritten in the following way. While all $(\boldsymbol{x},a)$ pairs are multiplied by $ \phi_{\boldsymbol{x},a}^* $,  $\eta_1$ in the numerator does  not affect final performance too much. The key is to find a good value of $\eta_1^2/\lambda$ to balance these two terms in the denominator:
>
> $$ \qquad \qquad \qquad \eta_1/\left[ \exp \left(-\gamma U_{\boldsymbol{x},a} \right) + \frac{ {\color{blue}\eta_1^2/\lambda} \cdot \pi_{\boldsymbol{\vartheta}}(a|\boldsymbol{x})^2}{\hat{\beta}(a|\boldsymbol{x})^2 \exp \left(-\gamma U_{\boldsymbol{x},a} \right)} \right].
> $$
> We have updated this part of the discussion in Appendix A.2 to enhance applicability.
>
> Recall that $\gamma$ is a function of $\delta$, where with probability at least $1-\sigma$,
> $|f_{\boldsymbol{\theta^*}}(\boldsymbol{x},a) -  f_{\boldsymbol{\theta}}(\boldsymbol{x},a)| \leq \gamma U_{\boldsymbol{x},a} $ holds. And typically the smaller the $\delta$ is, the larger the $\gamma$ is. Thus, to ensure the confidence of the calculated uncertainty, $\gamma$ should be relatively larger as shown in the ablation study on the effect of hyperparameters in Table 3.
>
> For $\eta_1^2/\lambda$, it is closely related to how UIPS works as discussed in "Insights on $\phi_{\boldsymbol{x},a}^*$" in Section 3.1. When the propensity score $\frac{\pi(a|\boldsymbol{x})}{\hat{\beta}(a|\boldsymbol{x})}$ is larger than $\sqrt{\lambda}/\eta_1$, UIPS will assign a smaller weight to a sample with more uncertain $\hat{\beta}(a|x)$ to prevent its distortion. Thus $\sqrt{\lambda}/\eta_1$ should be larger in problems with larger action spaces to prevent being too conservative.
>
> We select the hyperparameters based on the model’s performance on the unbiased validation dataset. The range of hyperparameter search can be found in Implementation Details in Appendix A.2.
>
> **Construction of the unbiased validation datasets.** In our experiments, since feedback on each action is available on our synthetic datasets, we randomly sample 3K samples in the dataset as the validation dataset. On the three real-world datasets, each contains a set of biased data collected from users' interactions on the platform for training and a set of unbiased data collected from a randomized controlled trial where items are randomly selected. We used a small part of the unbiased data as the validation dataset.
>
> In real-world scenarios, especially in industrial recommender systems, one can always get the unbiased validation dataset via a small-traffic randomized controlled trial. Plenty of previous work [1,2,3] used a similar approach to obtain an unbiased dataset for model learning in practice.
>
> [1]Chen, Minmin, et al. "Top-k off-policy correction for a REINFORCE recommender system." WSDM 2019.
>
> [2]Ma, Jiaqi, et al. "Off-policy learning in two-stage recommender systems." Proceedings of The Web Conference 2020. 2020.
>
> [3] Chen, Jiawei, et al. "AutoDebias: Learning to debias for recommendation." SIGIR2021.

---

### Official Review · Reviewer_DgbC · 2022-10-25

**Confidence:** 3
**Correctness:** 4
**Technical Novelty And Significance:** 3
**Empirical Novelty And Significance:** 3
**Recommendation:** 6

**Clarity, Quality, Novelty And Reproducibility:**

A high quality, well written paper.
As far as I can tell, the work presented is novel, and all code for reproducing results was submitted.

**Strength And Weaknesses:**

Strengths:
- The paper is clearly written and well organized. It was pleasant to read, and the reasoning flowed nicely.
- The paper tackles a common problem that's encountered across several industries, how to apply off-policy evaluation when the logging policy is unknown. The typical approach of estimating a logging policy from data suffers from high bias, and this paper addresses that issue by including uncertainty when weighting samples with probability ratios.
- The experimental results show that UIPS consistently outperforms the chose benchmark techniques, when the logging policy is estimated.


Weakness:
- I would have liked to see a comparison in the synthetic dataset of how close the proposed technique gets to the ground-truth logging policy, when it is estimated. That oracle baseline would be very helpful to understand how significant the improvements of UIPS are in contrast to the performance of the other methods.
For ex. if UIPS achieves P@5 of 0.5666 (table 1), the difference to CE 0.5559 is a lot more meaningful if the performance of using the true logging policy at P@5 is 0.5700, than if it is 0.988.

- In my experience, doubly-robust (DR) estimators are often some of the best performing benchmarks for off-policy evaluation. I would have liked to see the use of a DR estimator as part of the evaluations.

- For experimental results, please include standard deviation numbers as well as mean performance as it helps to identify how significant those improvements are.


Nitpick:
- Page 8, section 4.1, The second sentence makes reference to Table 3 showing the average MSE, but table 3 is actually showing P@5, R@5. Maybe a wrong table reference, or something got lost in editing?

- Table 2: Any comment on why would CE outperform other off-policy eval methods on low frequency actions? I'm very intrigued by that, as I would expect CE to be heavily biased specially in low frequency samples.

**Summary Of The Paper:**

This paper proposes an extension to the classical inverse propensity score estimator for off-policy learning.
The authors propose considering the uncertainty in estimation to the logging policy when this policy is unknown, which is typically the case in many practical scenarios. They do so by introducing a new learnable parameter ( phi ), that increases for samples with low uncertainty, and decreases for samples with high uncertainty.
Through various datasets, the authors show that their proposed technique is able to outperform several standard methods for off-policy evaluation.


**Summary Of The Review:**

A well-written paper that highlights a practical issue of estimating logging policies when these are not available.
This paper tackles the problem by taking into account uncertainty in estimating the logging policy, and making corrections to the classical propensity scores based on uncertainty.
Throughout several experiments on a number of datasets, the authors show that the propose technique improves results over other commonly used techniques.

---

> ### Author Response · Authors · 2022-11-17
> **Response to Reviewer DgbC**
>
> We thank the reviewer for the positive comments on our work and valuable suggestions on the experiment design. Based on the suggestions, we have included the suggested baseline with the ground-truth logging probabilities on synthetic datasets, experiments on DR estimators, and reported standard deviation numbers of all experimental results under 10 random seeds.
>
> [Q1]**"I would have liked to see a comparison in the synthetic dataset of how close the proposed technique gets to the ground-truth logging policy, when it is estimated."**
>
> [A1] We have included a new baseline IPS-GT, which depicts the performance an IPS estimator can achieve, assuming the ground-truth logging probabilities are known and sample size is sufficiently large, as shown in **Table 1 in the updated paper**.
>
> We can observe that UIPS achieved similar and even better performance than IPS-GT when $\tau=0.5$ and $\tau=1$, but performed worse than IPS-GT on the dataset with $\tau=2$.
> Although IPS-GT can access the ground-truth logging probabilities, it still suffers from high variance caused by samples with small logging probabilities, leading to the worse performance  when $\tau=0.5$ and $\tau=1$, while UIPS can control the negative impact from these samples since they are usually associated with high uncertainties.
>
> When the ground-truth logging policy is smoother (e.g., $\tau=2$), the variance of the IPS estimator becomes much smaller, and off-policy correction with the ground-truth logging probabilities, rather than the estimated ones, leads to better model performance.  But UIPS still outperforms all baselines with an unknown ground-truth logging policy.
>
> [Q2]**"I would have liked to see the use of a DR estimator as part of the evaluations."**
>
> [A2]  Following the reviewer's suggestion, we inspected the performance of the DR estimator on all datasets. We also integrated the proposed UIPS to the doubly-robust settings with the same imputation model, named UIPSDR. The imputation model is pre-trained following the previous work [1].  The experiment results are reported in **Table 8 (the synthetic datasets) and Table 9 (three real-world datasets) in Appendix A.3 in the updated paper**.
>
>
> We observed that **DR cannot consistently outperform IPS-Cap**: It outperforms IPS-Cap on the Coat and KuaiRec dataset, while achieving much worse performance on the synthetic datasets and Yahoo dataset. This is because **the imputation model also plays an important role in gradient calculation as shown in Equation(13) in Appendix A.3, thus its accuracy greatly affects the policy learning**.  When the imputation model is sufficiently accurate, for example on the Coat dataset with only 300 actions, incorporating the DM estimator not only leads to better performance of DR over IPS-Cap, but also improved performance of UIPSDR over UIPS. And in particular, in this situation, UIPSDR performs better than DR with the gain being statistically significant.
>
> When the imputation model is not accurate enough, for example, on the KuaiRec dataset with a large action space but sparse reward feedback, DR is still worse than UIPS, and UIPSDR also performs worse than UIPS, due to the distortion of the imputation model.
>
> [1] Liu, Yaxu, et al. "Practical Counterfactual Policy Learning for Top-K Recommendations." KDD2022.
>
>
> [Q3]**"For experimental results, please include standard deviation numbers as well as mean performance as it helps to identify how significant those improvements are."**
>
> [A3] Please find our answer to CQ1 in the general response to all reviewers.
>
> [Q4] **"Any comment on why would CE outperform other off-policy eval methods on low frequency actions? "**
>
> [A4] It is because the estimated logging probabilities of low-frequency actions are very likely to be inaccurate (i.e., their estimation uncertainty is high), which dramatically distorts the quality of offline learning on them, leading to the performance even worse than CE. On the contrary, when the estimated logging probabilities become more accurate, i.e., on those high-frequency actions, the advantage of baseline offline learning algorithms emerges as shown in Table 2.

---

> > ### Comment · Reviewer_DgbC · 2022-11-23
> > **Follow up from response**
> >
> > I would like to thank the author for the time and effort they put in addressing my comments.
> > This will be taken into consideration when providing a final score.

---

> > > ### Author Response · Authors · 2022-11-23
> > > **We would like to get engaged in the discussion**
> > >
> > > Dear reviewer,
> > >
> > > We noticed the overall recommendation has recently been downgraded from 8 to 6 after the rebuttal period.
> > >
> > > We would like to follow up with the reviewer and get engaged in the discussions to provide any necessary inputs and clarifications. We just do not want any possibly misunderstandings or minor issues affect the outcome of our scientific discovery.
> > >
> > > Thank you all for your input and help in improving the quality of this submission.

---

### Official Review · Reviewer_FTLw · 2022-10-26

**Confidence:** 3
**Correctness:** 3
**Technical Novelty And Significance:** 3
**Empirical Novelty And Significance:** 3
**Recommendation:** 8

**Clarity, Quality, Novelty And Reproducibility:**

**Clarity:**
The paper is somewhat clearly written, although there is a lot of mathematical notation to remember, which makes the paper harder to understand. In addition, there are frequent and numerous grammatical errors.

**Quality:**
I don't have any major complaints about the quality of the paper.

**Novelty:**
To the best of my knowledge, the proposed algorithm is novel, but I am also not up to date on the latest literature in this area.

**Reproducibility:**
The paper provides experimental details in an appendix, and should be reproducible without too much work.

**Strength And Weaknesses:**

**Strengths:**
+ The paper draws on work from several areas to build the proposed algorithm.
+ It seems like future algorithms for uncertainty estimation in neural networks could be dropped into the proposed method with only minor modifications, making the algorithm somewhat future-proof in this respect.
+ Strong performance in experiments.

**Potential Weaknesses/Questions:**
- Some of the citations do not seem to cite the correct papers. For example, the earliest use of probability ratios in off-policy RL that I'm aware of is Sutton and Barto (1998), and the earliest peer reviewed paper using probability ratios in off-policy RL that I'm aware of is Precup et al. (2000).
- The title of the paper seems overly broad, as the uncertainty is only in the estimated logging policy, not in the resulting policy or value estimates, and the proposed algorithm is only for the contextual bandit setting.
- ~~The example of news recommendation at the beginning of Section 2 seems a little weird. If the context summarizes the user's interaction with the recommender system, then their choice of action would affect future contexts, which would be reinforcement learning and not a contextual bandit problem. If that's true, why use the contextual bandit setting?~~
- It seems like the proposed algorithm is limited to the finite action case, and does not support continuous actions spaces, or generalization between actions.
- ~~Can the authors please comment on the computational complexity of the proposed method in comparison to the competitor methods used in the experiments? Does UIPS use more computation than the other methods?~~
- It would be good to explicitly specify the algorithm somewhere self-contained (even in an appendix would be fine if space is a concern), to make it easier to reproduce and easier for practitioners to implement.
- There's a lot of notation to remember which makes the paper harder to understand, and some of it is not necessary. For example, on page 5, why define the function $g(x_n,a_n)$ as the gradient of $f_\theta(x_n,a_n)$ instead of just writing $\nabla_\theta f_\theta(x_n,a_n)$?

**Minor comments/questions:**
- Page 3: "enlarges the this bias" has a typo.

**References:**
* Sutton, R. S., & Barto, A. G. (1998). Reinforcement learning: An Introduction.
* Precup, D., Sutton, R. S., & Singh, S. P. (2000). Eligibility Traces for Off-Policy Policy Evaluation. ICML.

**Summary Of The Paper:**

The paper is concerned with accurately estimating the logging policy in off-policy learning when it is unknown or unavailable. Specifically, the paper derives an uncertainty-aware inverse propensity score algorithm called UIPS that is more accurate for rarely-taken actions, and provides theoretical and empirical arguments in favour of the algorithm.

**Summary Of The Review:**

I am conflicted. The proposed algorithm is interesting and performs well in experiments, but my concerns with clarity and readability prevent me from wholeheartedly recommending acceptance. If the authors address my concerns, improve the clarity of the paper, and drastically reduce the grammatical errors, I would recommend acceptance.

**Update:** The authors have addressed enough of my concerns that I now recommend accepting the paper for publication.

---

> ### Author Response · Authors · 2022-11-17
> **Response to Reviewer FTLw**
>
> We thank the reviewer for the constructive suggestions to clarify the arguments in our work and improve readability of the paper.
>
> Following the reviewer’s suggestion, we improved the clarity of the paper through several rounds of proofreading, summarized important notations in Table 6 and the whole algorithm framework in Algorithm 1 in Appendix A.1 (all marked in blue), and added suggested citations, etc. For other questions, please refer to our responses as follows.
>
> [Q1] **"The example of news recommendation at the beginning of Section 2 seems a little weird. If the context summarizes the user's interaction with the recommender system, then their choice of action would affect future contexts, which would be reinforcement learning and not a contextual bandit problem. If that's true, why use the contextual bandit setting?"**
>
> [A1] We adopted the contextual bandit setting to study the off-policy learning problem and our proposed method, since contextual bandits make no assumption about how the contexts are generated or evolved across decision rounds, i.e., the contexts can be dependent, independent, or even adversarially generated based on users' previous interactions [5,6]. Therefore, it is a more general formulation for the interactive recommendation problems. Moreover, the contextual bandit setting has been widely adopted in previous work on off-policy learning in recommender systems [1,2].
>
> On the other hand, reinforcement learning typically assumes the learning problem follows a Markov Decision Process, which imposes a stronger environment assumption than the contextual bandit. We do agree with the reviewer that when the environment follows the MDP assumption, the reinforcement learning can be more advantageous than contextual bandit algorithms. The discussion about which is a better algorithm for recommender systems is beyond the scope of this work. According to Achiam et al. [4], when the environment follows the MDP assumption, the impact of applying off-policy learning in contextual bandit setting on the total reward of learned policy is bounded. Hence, it is still a viable choice of model for learning in such an environment.
>
> [1]Chen, Minmin, et al. "Top-k off-policy correction for a REINFORCE recommender system." WSDM 2019.
>
> [2]Ma, Jiaqi, et al. "Off-policy learning in two-stage recommender systems." Proceedings of The Web Conference 2020. 2020.
>
> [3]Li, Lihong, Rémi Munos, and Csaba Szepesvári. "Toward minimax off-policy value estimation." Artificial Intelligence and Statistics. PMLR, 2015
>
> [4]Achiam, Joshua, et al. "Constrained policy optimization." International conference on machine learning. PMLR, 2017
>
> [5] Abbasi-Yadkori, Yasin, Dávid Pál, and Csaba Szepesvári. "Improved algorithms for linear stochastic bandits." Advances in neural information processing systems 24 (2011).
>
> [6] Li, Lihong, et al. "A contextual-bandit approach to personalized news article recommendation." Proceedings of the 19th international conference on World wide web. 2010.
>
> [Q2] **"It seems like the proposed algorithm is limited to the finite action case, and does not support continuous actions spaces, or generalization between actions."**
>
> [A2] Yes, this paper mainly focuses on the finite and discrete action space, which has been the main focus of the community in this line of research. Previous work with continuous action space [7] also used a similar component as the propensity model to achieve the goal of off-policy learning. Hence, it paves the way for us to apply our method to the continuous action problems. We thank the reviewer for pointing out this interesting extension to continuous action space and generalization between actions, and we will consider it as an important direction of our future work.
>
> [7] Chernozhukov, Victor, et al. "Semi-parametric efficient policy learning with continuous actions." Advances in Neural Information Processing Systems 32 (2019).
>
> [Q3] **"Can the authors please comment on the computational complexity of the proposed method in comparison to the competitor methods used in the experiments? Does UIPS use more computation than the other methods?"**
>
> [A3] Please find our answer to CQ2 in the general response to all reviewers.
>
> [Q4] **"There's a lot of notation to remember which makes the paper harder to understand, and some of it is not necessary. "**
>
> [A4] To help readers better follow our algorithm design, we summarize all notations used in our paper in Table 6 of Appendix A.1, following the convention in [8].
>
> [8] Zhou, Dongruo, Lihong Li, and Quanquan Gu. "Neural contextual bandits with ucb-based exploration." ICML2020.

---

### Author Response · Authors · 2022-11-17
**General response to the reviewers**

We thank all reviewers for their insightful comments and suggestions, which significantly helped us to strengthen our paper. We have uploaded a revised version of our submission where the major changes are highlighted in blue. In the following,  we will first respond to the common suggestions from all reviewers, and then respond to each reviewer individually.

[CQ1] **"Report standard deviation numbers as well as mean performance of experimental results."**

[CA1] We reported the average performance and standard deviations of all algorithms under 10 random seeds in **Table 1 (the synthetic datasets) and Table 5 (three real-world datasets) in the updated paper**. To show the significance of the improvements, the p-value under t-test between UIPS and the  best baseline on each dataset is also provided. The statistical results show that the improvement from UIPS against the baselines is significant. A short note is that we re-ran all experiments to report the standard deviation results; and therefore, the reported values are a bit different from our previous version of submission but the relative comparison among algorithms does not change.

[CQ2]**"The computational complexity of the proposed UIPS."**

[CA2] We summarized the details of our algorithm in Algorithm 1 in Appendix A.1, including its computational complexity. Given the logged dataset containing $N$ samples, $A$ actions and latent dimension $d$, the additional computation cost of UIPS over IPS comes from two parts (assuming one also needs to estimate the logging policy to use IPS, otherwise there is no need to use UIPS):

- Pre-calculating uncertainties (line 1-5 in Algorithm 1): This part calculates uncertainty of the logging probability for each sample, and **it only needs to be executed once**.  The computational cost of this step is  $O(Nd^2 + d^3)$, where $O(d^3)$ is for matrix inverse and $O(Nd^2)$ is for calculating uncertainties in each sample.
- Calculating $\phi_{\boldsymbol{x},a}^*$ during training (line 8 in Algorithm 1): It only takes O(1) time, the same computational cost as calculating IPS score.

Note that calculating logging probability for each sample, which is essential for both UIPS and IPS, takes $O(NAd)$ time. Since the dimension $d$ is usually much smaller than action size $A$ and sample size $N$, **UIPS does not introduce significant computational overhead compared to the original IPS**.

---

### Author Response · Authors · 2022-11-18
**A gentle reminder to the reviewers**

Dear reviewers,

Thank you again for your valuable time and thoughtful comments. We have provided detailed responses and additional experiment results to best answer the questions. As we are approaching the end of the discussion stage, we would appreciate it if you could read our responses and let us know if your concerns have been addressed. We are more than happy to further discuss any details that you find not fully addressed. Thank you.

Best regards,

Authors

---

### Decision · Program_Chairs · 2023-01-20

**Decision:**

Reject

**Justification For Why Not Higher Score:**

The paper is on a good topic but the algorithm design has issues.

This paper should be treated as a **weak reject**, reject and bump up if there are no better papers.

**Justification For Why Not Lower Score:**

This paper addresses an important practical problem and the empirical results are good.

**Metareview: Summary, Strengths And Weaknesses:**

This paper proposes uncertainty modeling for estimated logging policies in the IPS estimator. Specifically, the authors express the IPS estimator such that it involves the ratio of estimated and unknown logging policies, and then they overestimate it. The approach does not come with guarantees but it is comprehensively compared to 10+ baselines.

This paper had a wide range of scores, from accept to reject. All reviewers agree that the studied problem is novel, important in practice, and may motivate follow-up works. The most negative review is because the approach does not come with an error bound. While this may be concerning, I do not think that it would be a big issue if the algorithm design was clean and sound. The problem is that this is not clear. Here are a few things that came out during an online meeting with the reviewers:

* From Theorem 1 to (6), an unknown quantity is replaced with a constant $\lambda$. This constant needs to be properly bounded.

* Since (7) behave like logistic models, their confidence intervals should be chosen accordingly. This is unclear. The authors cite a linear bandit paper and their high-probability region does not seem additive in confidence interval widths. I would also expect something like one over the minimum derivative of the mean function term. See [Provably Optimal Algorithms for Generalized Linear Contextual Bandits](https://proceedings.mlr.press/v70/li17c.html) and [Randomized Exploration in Generalized Linear Bandits](https://proceedings.mlr.press/v108/kveton20a.html) for proper confidence intervals in related generalized linear models.

* The algorithm seems complicated. I say "seems" because it is never precisely stated. Specifically, what is the advantage of the upper bound in Theorem 2 over having a high-probability lower bound on $\beta^*(a \mid x)$ in an IPS estimator with $\beta^*(a \mid x)$? This one could be derived using the confidence interval on page 5, no?

Given the above concerns, no reviewer felt strongly about accepting the paper. It is a borderline with a bias to reject.

**Summary Of Ac-Reviewer Meeting:**

All points in the meta-review were discussed. The algorithmic issues arose during the discussion. At the end, the most positive reviewer agreed that they are fine with a weak reject, reject and bump up if there are no better papers.